**EMBO** *reports*

# Endocytosed lipids induce cell aggregation via filopodia retraction in a close relative of animals

Ria Q Kidner [1,3], Eleanor B Goldstone[1], Henry J Rodefeld[1], Lorin P Brokaw[1], Aria M Gonzalez[1], Lalitha Sastry[1], Ranojoy Baisya [1], Núria Ros-Rocher [2,4] & Joseph P Gerdt [1✉]

## Abstract

*Capsaspora owczarzaki* **is a protist that may both reveal aspects of animal evolution and curtail the spread of schistosomiasis.** *Capsaspora* **exhibits a regulated aggregative behavior reminiscent of cellular aggregation in some animals—a process that might have contributed to the origin of animals. This aggregative behavior may also be vital for** *Capsaspora* **to colonize the intermediate host of parasitic schistosomes and potentially prevent the spread of schistosomiasis. Both applications demand elucidation of the mechanisms underlying** *Capsaspora* **aggregation. Toward this goal, we evaluated the chemical properties of lipid cues that activate aggregation. We found that a range of zwitterionic lipids induced this behavior, revealing that aggregation is activated by diverse lipid-rich conditions. Furthermore, we demonstrated that aggregation in** *Capsaspora* **requires clathrin-mediated endocytosis, highlighting the potential significance of endocytosis-linked cellular signaling in recent animal ancestors. Finally, we found that aggregation is initiated independently of protein translation, suggesting post-translational activation of filopodial retraction. Together, our findings illuminate the molecular and cellular basis of** *Capsaspora*'**s aggregative behavior, with implications for the evolution of animal multicellularity and the transmission of parasites.**

**Keywords** Cellular Aggregation; Lipid Signaling; Endocytosis; Filopodia Dynamics; Multicellularity Evolution
**Subject Categories** Cell Adhesion, Polarity & Cytoskeleton; Evolution & Ecology

## Introduction

*Capsaspora owczarzaki* (hereafter "*Capsaspora*") is a close relative of animals of particular interest for both the origins of animal multicellularity and the parasitic disease schistosomiasis (Ferrer-Bonet and Ruiz-Trillo, 2017). *Capsaspora's* genome encodes genes homologous to those critical for animal cell–cell signaling and adhesion, among others (Sebé-Pedrós et al, 2016a; Suga et al, 2013). Notably, many of these genes are upregulated during its reversible aggregative stage (Bercedo-Saborido et al, 2025; Li et al, 2025; Sebé-Pedrós et al, 2013). In animals, reversible aggregation is a fundamental process involved in development, tissue maintenance, and immune responses. For example, aggregation enables whole body regeneration in sponges and hydras (Ereskovsky et al, 2021; Gierer et al, 1972), facilitates cellular ingression during gastrulation of mammalian epiblasts (Simunovic et al, 2019), and supports migration of neuronal cells in developing nervous systems (McKeown et al, 2013) and immune cells during inflammation (O'Flaherty and Ward, 1978). Thus, cellular aggregation may have been an important part of the life cycle of the first multicellular animals (Brunet and King, 2017; Ros-Rocher et al, 2021; Ruiz-Trillo et al, 2023). Despite this, it is still unclear which aspects of reversible aggregation observed in present-day animals are homologous to the morphologically similar behaviors seen in *Capsaspora* (Ros-Rocher et al, 2021; Sebé-Pedrós et al, 2013; Suga et al, 2013) or in other unicellular animal relatives (Brunet and King, 2017; Li et al, 2025; Ros-Rocher et al, 2026; Woznica et al, 2017).

In addition to its evolutionary significance, *Capsaspora* has potential biomedical importance. It was originally isolated from the snails that serve as intermediate hosts for parasitic schistosome worms that cause schistosomiasis (Hertel et al, 2002; Stibbs et al, 1979). *Capsaspora* can sense these parasites and kill them in their intramolluscan life stage, potentially preventing transmission to humans (Owczarzak et al, 1980; Quick et al, 2025; Stibbs et al, 1979). Moreover, *Capsaspora* responds to specific snail-derived lipids (Kidner et al, 2024; Ros-Rocher et al, 2021) by forming large aggregates (both in vitro and within snail tissue), suggesting that this behavior is important for host colonization. However, we still poorly understand the regulatory mechanisms by which *Capsaspora* senses these host-derived factors.

For both reasons, we sought to elucidate the molecular and cellular mechanisms that regulate multicellular aggregation in *Capsaspora*. Although previous studies have made significant progress in characterizing some aspects of this phenomenon (Ros-Rocher et al, 2021; Sebé-Pedrós et al, 2013; Sebé-Pedrós et al, 2016a; Sebé-Pedrós et al, 2016b; Suga et al, 2013), the

[1]Department of Chemistry, Indiana University, Bloomington, IN, USA. [2]Department of Functional Genomics and Evolution, Institut de Biologia Evolutiva (Consejo Superior de Investigaciones Científicas-Universitat Pompeu Fabra), Passeig Marítim de la Barceloneta 37-49, Barcelona, Spain. [3]Present address: Los Alamos National Laboratory, Los Alamos, NM, USA. [4]Present address: Department of Cell Biology and Infection and Department of Developmental and Stem Cell Biology, Institut Pasteur, Université Paris-Cité, CNRS UMR3691, 25-28 Rue du Docteur Roux, Paris, France. ✉E-mail: jpgerdt@iu.edu

mechanism(s) underlying the initiation and maintenance of aggregation in *Capsaspora* remain elusive. Building upon previous work demonstrating that low-density lipoproteins (LDLs), snail-derived lipids, and pure dioleoylphosphatidylcholine (DOPC) can induce *Capsaspora* aggregation (Kidner et al, 2024; Ros-Rocher et al, 2021), we first investigated whether other pure lipids could similarly induce aggregation. To identify the lipid properties necessary for aggregation, we tested a panel of natural lipids found in LDLs and snail serum, alongside several non-native lipids. We found that many zwitterionic diacyl lipids robustly induced aggregation. This suggests that *Capsaspora* (and potentially its ancestors) can sense a wide range of ubiquitous lipids, which could serve as general environmental cues associated with lipid-rich niches and/or the presence of nutritional prey.

Furthermore, we investigated the intracellular mechanism by which these lipid inducers trigger aggregation in *Capsaspora*. We found that endocytosis is necessary for aggregation, as chemical inhibition of multiple early steps in the endocytic pathway effectively blocked aggregate formation. In addition, we showed that aggregation is initiated within seconds of lipid addition and occurs independently of new protein synthesis. Finally, we found that cell–cell aggregation is driven by the rapid retraction of interconnected filopodia in all cells, leading to tight cell–cell contacts.

Taken together, our study sheds light on the lipid-mediated induction of cellular aggregation in *Capsaspora*, revealing key roles for endocytosis, post-translational regulation, and dynamic remodeling of filopodia. These findings not only advance our understanding of unicellular-to-multicellular transitions but also provide key insights into a behavior that may be important for a protective symbiosis with host snails.

## Results

### *Capsaspora* aggregation is induced by phosphocholine-containing diacyl lipids

In previous work, we found that fetal bovine serum (FBS), low-density lipoproteins (LDLs), snail serum, and pure dioleoyl phosphatidylcholine (DOPC) lipids induced robust cellular aggregation in *Capsaspora* (Kidner et al, 2024; Ros-Rocher et al, 2021). However, we suspected that additional natural lipids beyond DOPC could also trigger aggregation for a few reasons: (1) many lipids are structurally similar to DOPC, (2) the concentration of DOPC present in snail serum was insufficient to fully account for the aggregation-inducing activity of crude snail serum (Kidner et al, 2024), and (3) high-density lipoproteins (HDLs)—which contain only trace amounts of DOPC (Kontush et al, 2013)—also robustly induced aggregates (Figs. 1A,B and EV1). To systematically identify other phospholipids capable of inducing *Capsaspora* aggregation, we conducted a structure–activity relationship (SAR) analysis across a panel of common phospholipids.

Before starting the SAR study, we established a standardized method for lipid delivery by assembling the lipids into macromolecular vesicles. Lipids simply mixed with buffer and delivered to cells failed to induce aggregation, but even brief sonication formed soluble vesicles that reliably induced robust aggregation (Fig. EV2A) (Kidner et al, 2024). Thus, vesicle formation is essential for lipid

bioactivity, suggesting that *Capsaspora* recognizes its lipid cues in a vesicular context.

Next, we tested how the chemical composition of lipids within vesicles influences *Capsaspora* aggregation (Figs. 1C–K and EV3). Phospholipids are classically categorized based on the composition of their polar headgroups and lipophilic tails. Thus, we systematically varied these two components to assess their importance for aggregation induction, using the chemical structure of DOPC as a reference (Kidner et al, 2024) (Fig. 1C). Each lipid was tested across a range of concentrations (Fig. EV3), with the most active concentrations summarized in Fig. 1. To ensure that lipid vesicles were properly formed prior to biological assays, we characterized each lipid in the SAR study by transmission electron microscopy (TEM) (Fig. EV2B–Y).

Varying the structures of headgroups revealed that lipids with phosphatidylcholine (PC) headgroups were the most active at inducing aggregation compared to lipids with just a phosphate (i.e., phosphatidic acid, PA) or phosphatidyl ethanolamine (PE) (Fig. 1D,E). This indicated that the permanent positive charge afforded by the choline appears to be important for aggregation activity. The negative charge of the phosphodiester also appeared to be essential, as the matching ethylphosphocholine (EPC) lipid—which lacks this negative charge—failed to induce aggregation (Fig. 1D,E). Thus, these results indicate that *Capsaspora* aggregation is selectively triggered by zwitterionic lipids (i.e., lipids containing an equal number of positively and negatively charged functional groups). The importance of this zwitterion is further supported by the aggregation-inducing activity of sphingomyelin (SM) lipids (Fig. 1D,E). Intriguingly, not only the presence but also the positioning of the positive and negative charges on the zwitterionic head group was important: the lipid positive charge must be exterior to the negative charge, as evidenced by the inactivity of an inverted PC lipid containing an external negative charge (DOPCe) (Fig. 1D,E). The aggregation-inducing activity of SM lipids also suggested that the glycerol ester moieties of PCs can be modified without losing their effect. Indeed, we found that the stereochemistry of the glycerophosphocholine headgroup did not affect its activity (i.e., the enantiomeric PC still induced aggregation) (Fig. 1D,E). Furthermore, a diether PC analog was active. Together, these findings indicate that to induce aggregation in *Capsaspora*, lipids must possess a zwitterionic headgroup with the positive charge facing outward, but the glycerol core can be modified.

We next determined which features of the phospholipid tails are necessary for *Capsaspora* aggregation. Molecules lacking lipophilic tails—such as choline and phosphocholine—were inactive, indicating that the presence of lipophilic tails is essential to induce aggregation (Fig. 1F,G). In addition, variants of lysophosphatidylcholine (LPC) containing a single lipophilic tail were also inactive (Fig. 1G), suggesting that only diacyl zwitterionic lipids induce aggregation. Furthermore, at least one of the two tails must contain at least one degree of unsaturation to be active (Fig. 1H). However, there was no strict requirement for the geometry or location of the double bond. Lipids with double bonds further or closer to the headgroup retained activity, and lipids with either cis or trans olefins were equally active (Fig. 1I). These results suggest that *Capsaspora* aggregation is tolerant to a range of lipid tail configurations.

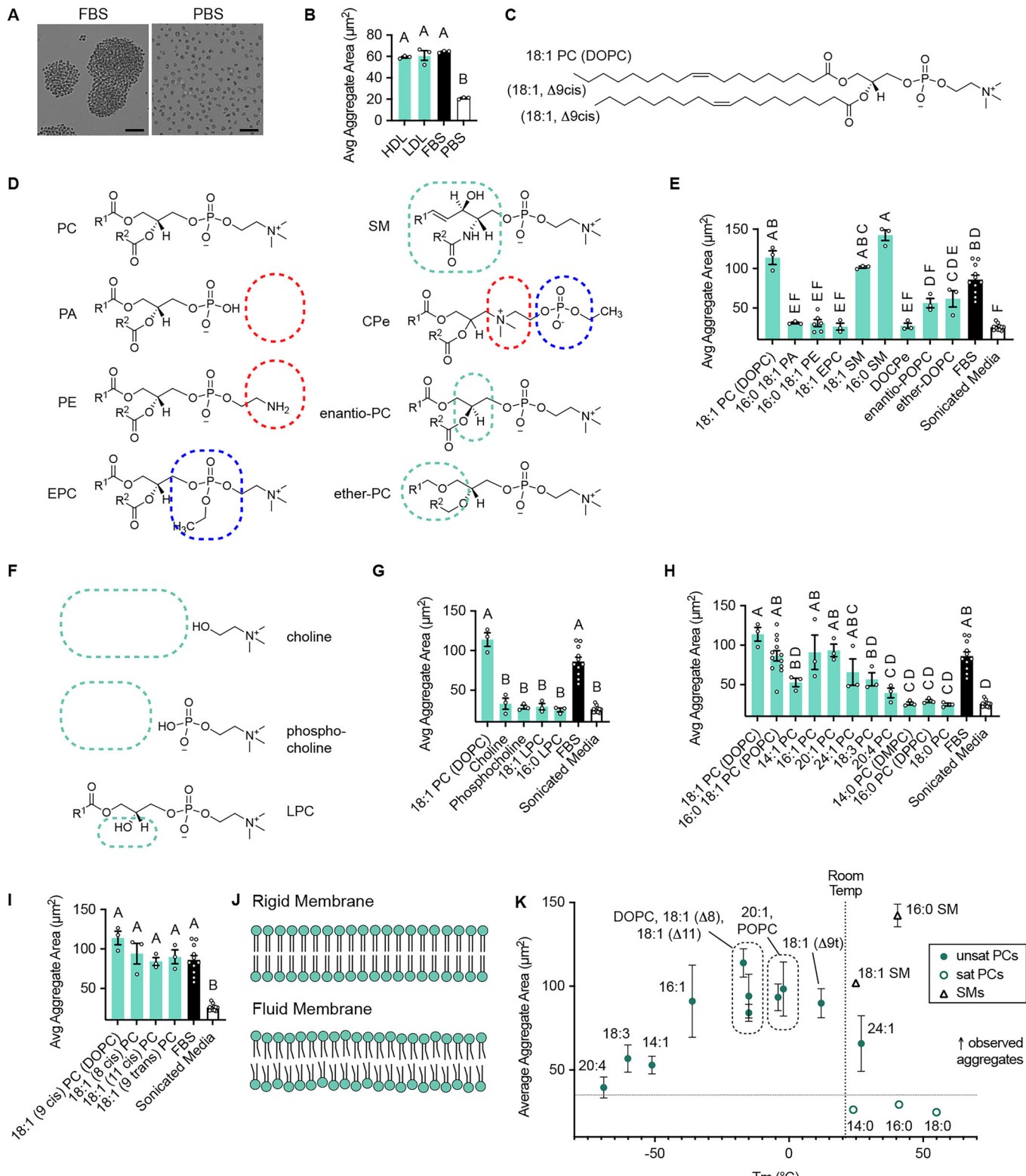

The requirement for unsaturation led us to consider the importance of lipid fluidity for inducing aggregation. Saturated lipids form more rigid lipid bilayers while unsaturated lipids form more fluid bilayers due to kinks introduced by double bonds (Fig. 1J). Since the gel phase transition temperature of phospholipids

correlates with fluidity (Marsh, 2013), we plotted all tested lipids in order of reported gel phase transition temperature values (Marsh, 2013) to determine if the physical fluidity of vesicles correlated with aggregation activity (Fig. 1K). Indeed, the most active phosphatidylcholine lipids had gel-phase transition temperatures below room

**Figure 1. Structure–activity relationship of lipids that induce aggregation in *Capsaspora*.**

(A) Representative images of *Capsaspora* cellular aggregates induced by 5% (v/v) FBS compared to single cells with 5% (v/v) PBS added. Scale bars are 20 μm. (B) Plot of average aggregate area induced by 80 μg/mL of high-density lipoproteins (HDL) and low-density lipoproteins (LDL), compared to controls 5% (v/v) FBS and 5% (v/v) PBS. HDL and LDL are active. See "Methods" and Fig. EV1 for more details on quantifying "Avg. Aggregate Area". Note that this metric includes single cells and tiny aggregates into the average. Therefore, the "average aggregate area" for a well is much smaller than the areas of the large aggregates in that well. (C) Chemical structure of DOPC, the previously reported active phospholipid (Kidner et al, 2024). (D) Head group structures for tested lipids. Dashed boxes indicate changes to the choline (red), the phosphodiester (blue), or the glycerol core (green). (E) Plot of aggregation-induction activity of lipids with varying head groups, showing that aggregation-induction activity is unique to phosphatidylcholine (PC) and sphingomyelin (SM) head groups. (F) Structures of choline molecules lacking lipophilic tails. Green dashed boxes indicate missing portions of the glycerol core and lipid tails relative to PCs. (G) Plot of aggregation-induction activity of molecules with different numbers of lipophilic tails, ranging from no tails to two tails. Phosphatidylcholine lipid monomers with two tails are active, while lyso-PCs and phosphocholine/choline alone are inactive. (H) Plots of aggregation resulting from PCs with different tail lengths and degrees of unsaturation. PCs with at least one degree of unsaturation were active, while no saturated PCs induced aggregation. (I) Plots of aggregation resulting from induction with PCs with different positions and stereochemistry of unsaturation. The geometry and location of the lipid tail double bond do not influence aggregate-inducing activity. (J) Cartoon of lipid membrane fluidity due to increased points of unsaturation. (K) Aggregation resulting from previously tested PCs and SMs plotted as a function of lipid gel phase transition temperature (Marsh, 2013). Activity of PC vesicles is generally associated with a gel phase transition temperature close to or below room temperature. In all aggregation bar plots, error bars represent the standard error of the mean ($n \geq 3$). Individual values are shown as white circles. For each inducer, multiple concentrations were tested (Fig. EV3), and the most active concentration is shown here. In all pure lipid aggregation-induction experiments, FBS was included as a positive control, and sonicated FBS-free media was used as a negative control. All panels with bar plots were analyzed by ordinary one-way ANOVA with a single pooled variance, using Tukey's multiple comparison test. Bars that share the same letter above them represent means that are not significantly different (corrected $P \geq 0.05$).

temperature (Fig. 1K). However, this correlation did not fully explain the observed results. For instance, sphingomyelins, which also induced robust aggregation, possess high gel-phase transition temperatures (e.g., SM(16:0) is ~40 °C), indicating that lipid fluidity alone cannot account for the full range of aggregation-inducing activity.

Overall, our findings define key structural requirements for lipids that induce *Capsaspora* aggregation: (1) delivery in the form of vesicles, (2) a zwitterionic headgroup with the positive charge exposed to the surface, and (3) two lipophilic tails, with at least one tail containing at least one degree of unsaturation.

## Initial cellular aggregation is activated post-translationally within seconds

We next turned to gain a deeper understanding of how *Capsaspora* responds to the aggregation-inducing lipids. In previous work, we observed aggregation initiated within three minutes of lipid addition, with aggregates gradually coalescing over the course of several hours before eventually dissipating (Ros-Rocher et al, 2021). To refine our understanding of the initiation dynamics, we performed higher-resolution time-series analyses and observed that cells start to aggregate within seconds after the addition of the inducer (Fig. 2A; Movie EV1). The timescale of this initial cell–cell aggregation is too rapid to be explained by changes in gene expression or de novo protein synthesis. We therefore hypothesized that the initial formation of multicellular aggregates does not require the expression of new proteins. To test this, we inhibited protein translation of non-aggregated cells using cycloheximide treatment and then tested if these non-translating cells could still aggregate in response to induction with FBS (containing both LDLs and HDLs). After confirming that cycloheximide blocked translation in *Capsaspora* (Fig. EV4), we found that cycloheximide did not inhibit aggregation (Fig. 2B). This finding demonstrates that lipid-induced aggregation of *Capsaspora* does not require translation of new proteins. Therefore, further investigation into the mechanism of *Capsaspora* aggregation response will require a focus on post-translational regulation.

## Induction of *Capsaspora* multicellular aggregates is dependent on clathrin-mediated endocytosis

In addition to the rapid induction of aggregation, we previously observed that *Capsaspora* internalizes and accumulates low-density lipoproteins (LDLs) within its cell body (Ros-Rocher et al, 2021). Given that LDL uptake in human cells occurs via clathrin-mediated endocytosis (Fig. 3A) (Goldstein et al, 1982), we hypothesized that *Capsaspora* may utilize a similar pathway for lipoprotein uptake. To test this, we first monitored the uptake of a fluorescently labeled LDL and found that it accumulated inside cells in small puncta (Fig. 3B), consistent with our previous observations (Ros-Rocher et al, 2021). We then doped a fluorescently labeled PC into pure DOPC vesicles and observed similar intracellular accumulation (Fig. 3C). To determine whether LDL is trafficked into acidic compartments, such as those found in endosomes and lysosomes, we used LDL labeled with the pH-dependent dye pHrodo red. The appearance of fluorescence inside cells suggested that LDLs accumulate in intracellular vesicles with low pH following uptake (Ritter et al, 2018) (Fig. 3D). Therefore, both LDL and PC particles are internalized by *Capsaspora*, likely through an endocytic route.

To further validate that lipid uptake occurs via bona fide endocytosis, we used pharmacological inhibition to block the endocytosis pathway (Fig. 3A). Specifically, we treated cells with Dynole 34-2, a dynamin inhibitor known to disrupt clathrin-mediated endocytosis (Hill et al, 2009). Dynole 34-2 significantly reduced the uptake of fluorescently labeled LDLs in a dose-dependent manner (Fig. 3E). Moreover, it inhibited aggregate formation in a dose-dependent manner, with similar $IC_{50}$ values ~5 μM (Fig. 3F). These results suggest that endocytosis of lipid particles is necessary for their aggregation-induction activity in *Capsaspora*.

To further explore this hypothesis, we then systematically blocked several steps of the endocytic pathway using a panel of well-characterized chemical inhibitors (Figs. 3A,G and EV5). Notably, blocking early and intermediate stages of endocytosis effectively prevented aggregation (Fig. 3A,G). For instance, inhibition of clathrin recruitment to the plasma membrane with Pitstop 2, or disruption of actin polymerization and branching with CK-666 (necessary for endosome budding into the clathrin pit),

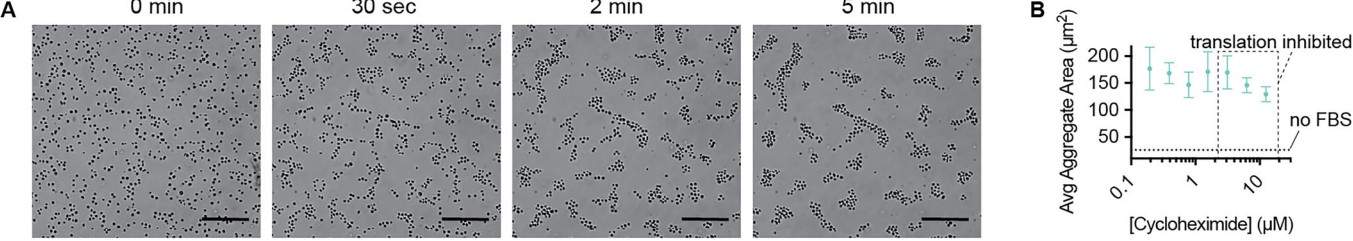

**Figure 2. *Capsaspora* aggregation starts within seconds after induction and does not require translation of new proteins.**

(A) Time series of brightfield images after induction by 5% (v/v) FBS. Scale bar is 50 μm. See Movie EV1. (B) Average aggregate area after 30-min pre-treatment with cycloheximide and induction with 5% (v/v) FBS. Error bars represent the standard error of the mean ($n = 3$).

prevented aggregation (Fig. 3G) (Hetrick et al, 2013). Similarly, inhibitors targeting dynamin recruitment (MiTMAB and OcT-MAB) and dynamin activity (Dynasore or Dynole 34-2) also prevented aggregation at concentrations comparable to those used in animal cell culture studies (Hill et al, 2009; Kirchhausen et al, 2008; Quan et al, 2007). Furthermore, inhibition of clathrin decoating from early endosomes using Chlorpromazine likewise prevented aggregate formation (Vercauteren et al, 2010). Finally, inhibiting endosome maturation from early to late endosomes with either Ponesimod or ABMA also prevented the formation of aggregates, suggesting that progression to the late endosome is required for the lipid signal to trigger aggregation (Fauzyah et al, 2021; Wu et al, 2017). Although these experiments were performed with FBS as the aggregation inducer, similar results were observed when we re-tested two of the inhibitors (Dynole and Dynasore) in conditions of aggregation induction by pure POPC vesicles (Fig. EV5L–N). Admittedly, each inhibitor has the potential for off-target effects that may inhibit aggregation apart from endocytosis blockade. Perhaps most notably, CK-666 may inhibit cell motility in a way that delays or prevents aggregation. However, the consistent inhibition of aggregation by this exhaustive set of inhibitors, each targeting a different component of the endocytosis pathway, suggests that endocytosis is required for aggregation.

In contrast, inhibition of later stages of endocytosis had little to no effect on preventing *Capsaspora* aggregation. For instance, blocking vesicle trafficking to endolysosomes using EGA (Gillespie et al, 2013)—even at very high concentrations—did not completely block aggregation. Moreso, inhibition of lysosome acidification using Monensin (Misinzo et al, 2008) failed to prevent aggregation (even though it did inhibit the formation of acidified endosomal compartments, Fig. EV5O). Taken together, these results strongly suggest that clathrin-mediated endocytosis is essential for lipid-induced aggregation in *Capsaspora*. The endocytosed LDLs (and presumably also the endocytosed PCs) ultimately accumulate in acidified lysosomes, as shown by activation of pHrodo dyes. However, our data suggest that the late stages of lysosome maturation are not required for initiating aggregation. Instead, only the early and intermediate steps of the endocytic pathway appear to be required to induce *Capsaspora* aggregation.

## Kinetics of aggregation correlate with phosphatidylcholine uptake

To further evaluate whether lipid endocytosis and aggregation are mechanistically linked, we compared their kinetics in real time. We

added fluorescently labeled DOPC vesicles to *Capsaspora* cells and simultaneously monitored cellular aggregation and lipid uptake. Aggregation began almost immediately following the addition of fluorescent PC-containing lipid particles (Fig. 4A–C; Movies EV2–4). In parallel, we also observed immediate binding of the fluorescent PC vesicles to the *Capsaspora* cell surfaces, particularly along the filopodia (Fig. 4A,B). The PC-stained filopodia revealed a striking network of filopodial contacts between cells. These filopodia connecting cells appeared to quickly retract, drawing cells together and leading to cellular aggregation within seconds of lipid exposure (Fig. 4A; Movies EV2–4). As aggregation proceeded, the fluorescent PCs appeared to traffic from the filopodia into the cell bodies, which became filled with bright puncta within minutes (Fig. 4C). By 4.5 min, the fluorescent signal along filopodia was still faintly visible in the PC channel, while the intracellular puncta in the cell bodies remained more prominent. These observations correlate with our earlier imaging after 30 min of incubation with fluorescent PC or LDL (Fig. 3B–D), which did not reveal substantial filopodial staining relative to puncta in the cell body. Overall, these observations suggest a tightly coupled sequence of events: lipid binding to filopodia, filopodial retraction, cellular aggregation, and internalization/trafficking of the lipid vesicles. The rapid onset of these processes further supports the hypothesis that lipid-induced aggregation is directly mediated by early steps in endocytosis.

In contrast, fluorescently labeled DPPC vesicles—which do not induce aggregation—did not adhere to the filopodia, and were not endocytosed (no white channel staining visible in Fig. 4D; Movie EV5). Therefore, only those lipids that induce aggregation (*e.g.*, DOPC) are bound and internalized by cells, again showing a clear correlation between lipid uptake and aggregation induction. These findings, together with our previous findings that endocytosis inhibitors blocked aggregation, suggest that *Capsaspora* binding and uptake of specific PCs is required for aggregation induction.

## Phosphatidylcholine vesicles are trafficked along filopodia to the cell body

Since the fluorescent PC liposomes were initially localized in the filopodia and then accumulated as puncta in the cell bodies, we hypothesized that they were trafficked along the filopodia toward the cell body. However, since the filopodia were not independently stained, we could not rule out the possibility that liposomes were

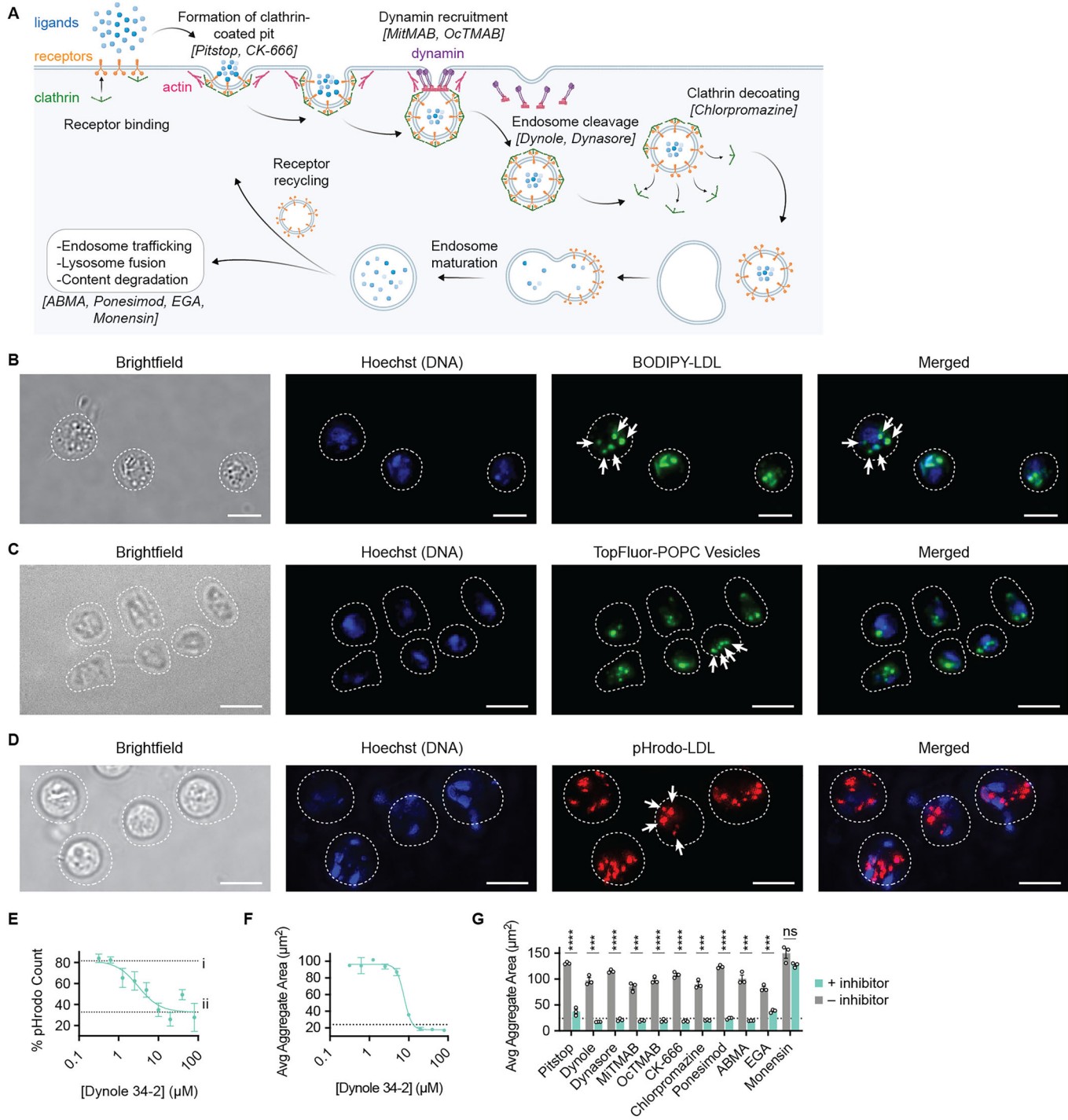

fixed onto the filopodia and pulled inward by retraction of filopodia to which they were bound. To distinguish between these possibilities (i.e., active trafficking versus passive retraction along with the filopodia), we added fluorescent DOPC vesicles to cells that were membrane-labeled with an N-terminal myristylation motif (NMM)-mVenus fusion (Fig. 5A; Movies EV6–9) (Phillips and Pan, 2024). Although the membrane and PC fluorescence channels overlapped, we observed instances of bright PC puncta

appearing to traffic along extended, non-retracting filopodia into the cell bodies. Furthermore, this uptake and trafficking were prevented by the endocytosis inhibitor Dynole (Movies EV10 and 11). However, this trafficking was not universal: many vesicles appeared to be stuck on filopodia and failed to traffic (Movies EV6–9). Since the event sometimes occurs, we hypothesize that *Capsaspora* can actively traffic exogenously added PC vesicles along outstretched filopodia (Fig. 5B), but yet-unknown factors

**Figure 3. Aggregation is dependent on clathrin-mediated endocytosis of LDL and POPC.**

(A) Stages of clathrin-mediated endocytosis, with the pharmacological inhibitors used to target each stage labeled in brackets. Panel (A) was created with BioRender.com. (B) Cellular uptake of BODIPY-labeled LDL shows accumulation of fluorescent puncta in adherent cells (white arrows). (C) Uptake of POPC doped at 5% (w/w) with TopFluor-PC shows similar results to LDL and accumulation of puncta inside adherent cells (white arrows). (D) Uptake of pHrodo-LDL shows accumulation of red fluorescent puncta in acidified endosomes/lysosomes (white arrows). Dashed lines in (B–D) indicate cell bodies; scale bars are 5 μm. (E) Percent of adherent cells with pHrodo red fluorescence after inhibition of LDL uptake with the dynamin inhibitor Dynole 34-2. The inhibitor blocked fluorescent pHrodo LDL accumulation. Label (i) is the percent of cells with pHrodo staining in the absence of Dynole. Label (ii) is the background % of fluorescence that appears as pHrodo staining in the absence of pHrodo addition. (F) Average aggregate area of cells treated with a dilution series of the inhibitor Dynole 34-2. Aggregation was induced by 5% FBS (v/v) 30 min after inhibitor treatment. The dynamin inhibitor Dynole 34-2 inhibited aggregation with a potency that correlated with its inhibition of LDL uptake. The dashed line indicates the baseline size of non-aggregated cultures. (G) Average aggregate area upon treatment with various chemical inhibitors along the clathrin-mediated endocytosis pathway. The following concentrations are shown: Pitstop (160 μM), Dynole (80 μM), Dynasore (60 μM), MiTMAB (160 μM), OcTMAB (10 μM), CK-666 (187 μM), Chlorpromazine (110 μM), Ponesimod (74 μM), ABMA (397 μM), EGA (144 μM), and Monensin (42 μM). See Fig. EV5 for complete dose response curves. Aggregates were induced with 5% FBS (v/v) 30 min after treatment with each inhibitor. Gray bars indicate DMSO controls tested concurrently with the matching inhibitor treatment. Consistent aggregation inhibition along the pathway shows that the initial stages of endocytosis are necessary for aggregation. In all plots, error bars represent standard error of the mean of a biological triplicate ($n = 3$). Individual biological replicates are displayed with white circles, and each treatment was compared with its own untreated control by a $t$ test. $P$ values for each comparison follow: Pitstop ($< 0.0001$), Dynole (0.0001), Dynasore ($< 0.0001$), MitMAB (0.0002), OcTMAB ($< 0.0001$), CK-666 ($< 0.0001$), Chlorpromazine (0.0001), Ponesimod ($< 0.0001$), ABMA (0.0003), EGA (0.0003), Monensin (0.0848).

may determine the likelihood of this process. Improved imaging that maintains filopodia in the focal plane and labels filopodia and lipids in separate channels will enable robust and quantitative assessments of this putative trafficking.

## Cells coalesce via retraction of interconnected filopodia, but non-connected filopodia can remain outstretched

Our earlier observation that PC-labeled filopodia appeared to retract (Fig. 4) relied on using fluorescent DOPC vesicles to indirectly "stain" the filopodia. However, this approach has two key limitations: (1) the mechanism of labeling is still unclear, so we are not certain if all filopodia are 'stained' uniformly by fluorescent PCs without bias, and (2) the fluorescence signal is quickly lost from filopodia as PC vesicles are trafficked into the cell body, making it difficult to monitor filopodia dynamics over time. The second point was particularly problematic for addressing a new question: *do all filopodia retract upon PC stimulation, or is filopodial retraction restricted to those engaged in cell–cell contact?* To more directly and reliably visualize filopodia, we used cells expressing the genetically encoded membrane marker NMM-mVenus (Phillips and Pan, 2024).

To assess the retraction of filopodia, we then added unlabeled PC vesicles and monitored the movement of membrane-labeled filopodia (Fig. 5C,D; Movies EV12–16). After PC addition, filopodia of neighboring cells appeared to connect and pull the cells together (filled white arrows). This finding strongly supports our hypothesis that filopodial retraction coalesces cells into aggregates. Surprisingly, in these aggregating cells with retracting filopodia, not every filopodium retracted. Some filopodia that were not contacting other cells remained extended (dashed outline arrows). Therefore, the PC lipids did not uniformly induce retraction of all filopodia throughout the cell. This observation suggests that PCs may not be sufficient to induce filopodia retraction, but retraction may also require cell–cell contact by a filopodia as a *second 'signal'* for that filopodium to retract (Fig. 5E). Filopodia are highly dynamic cellular components that are also used for substrate adhesion and motility (Parra-Acero et al, 2020); therefore, PCs may have important influences on these filopodia-driven behaviors, as well.

# Discussion

We discovered that *Capsaspora* aggregation is post-translationally induced by an assortment of zwitterionic diacyl phospholipids. We also found that the lipids are incorporated into cells through clathrin-mediated endocytosis, a multi-step process that is necessary for the lipids to induce *Capsaspora* aggregation. Furthermore, we revealed that aggregation of cells occurred through post-translationally inducing the retraction of interconnected filopodia, a process that required both uptake of inducing lipids and cell–cell filopodial contacts to initiate.

In our previous work, we observed that *Capsaspora* aggregation was induced by DOPC (Kidner et al, 2024). Here, we expand on that finding by showing that *Capsaspora* aggregates in response to all tested phosphatidylcholine (PC) lipids containing at least one unsaturation in at least one lipid tail. We also found that sphingomyelins, which share similar headgroup properties, were also potent inducers of aggregation. This promiscuous activation of aggregation suggests that *Capsaspora* (or its ancestors) may have evolved to aggregate in response to lipid-based cues associated to a variety of nutritional prey, not just cues from only snail hosts. As an osmotrophic feeder, *Capsaspora* likely secretes digestive enzymes to release and absorb nutrients from prey. The aggregation of many cells around a common food source—such as a prey cell, colony, or tissue—would maximize the efficiency of nutrient acquisition by allowing the cells to efficiently concentrate digestive enzymes and share released nutrients as common goods at a localized area (Fig. 6A) (Darch et al, 2012). Zwitterionic lipids are universally abundant in the membranes of animals (mainly PCs), bacteria (mainly PEs), fungi (PCs and PEs), and algae (PCs). Since these lipids are frequently released in extracellular vesicles by living cells (Skotland et al, 2020) and by lysing cells (Koga and Kusaka, 1968), they would be excellent cues to signal the presence of prey. Therefore, while DOPC and other PCs present in snail serum may serve as aggregation cues for *Capsaspora* in response to its snail host environment (Kidner et al, 2024), our findings suggest that this aggregation behavior may also be relevant outside of the *Capsaspora*–snail symbiosis. Specifically, it could help *Capsaspora* to colonize other, as-yet-unknown animal hosts or to form cooperative feeding aggregates around colonies of other prey

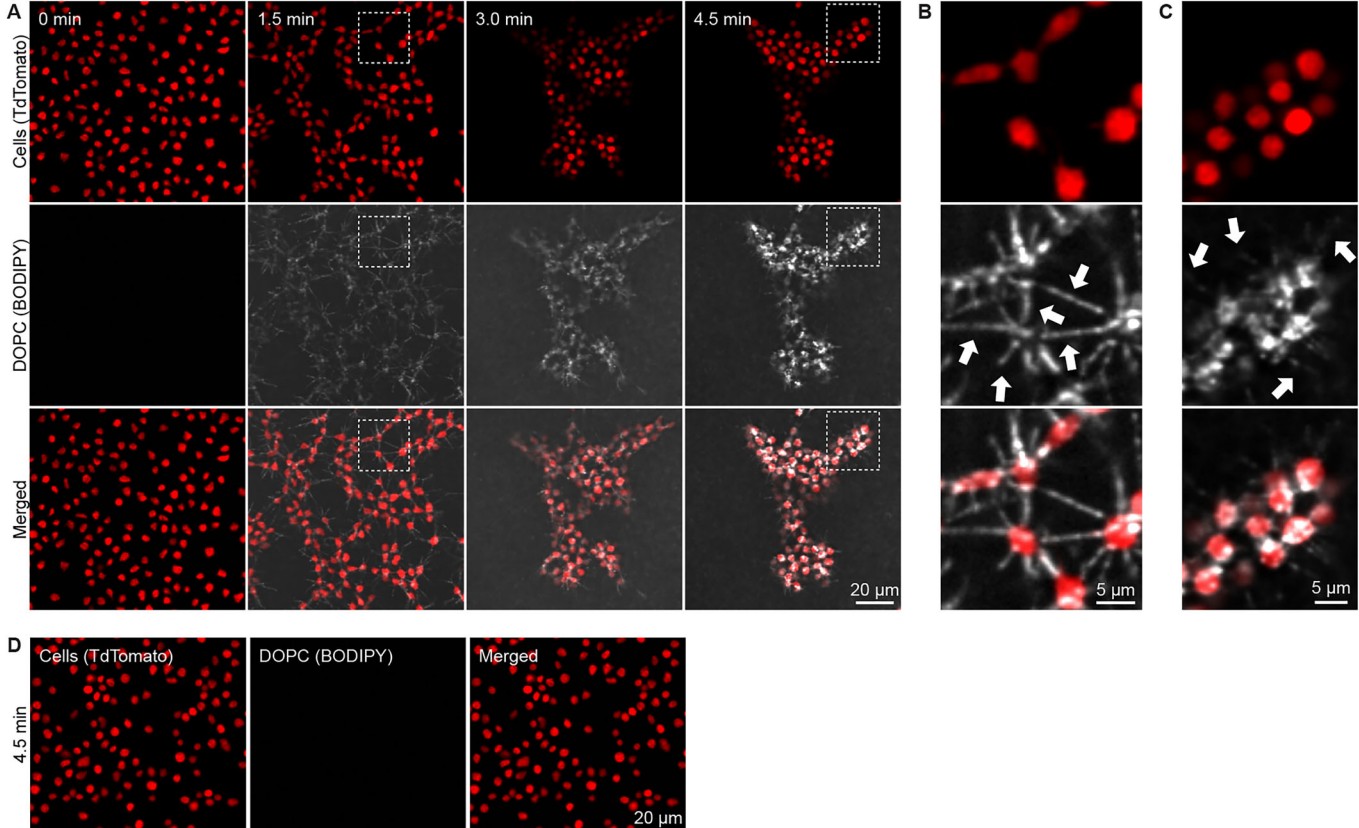

**Figure 4. Unsaturated PCs are taken up by filopodia and trafficked to the cell body with similar kinetics as aggregation.**

(A) Time series of fluorescence microscopy images after addition of fluorescent PC particles. The red channel shows TdTomato-expressing *Capsaspora* cells. The white channel shows fluorescent PC particles (DOPC/TopFluorPC [20:1 (w/w)]). Fluorescent PCs are rapidly taken up, correlating with cells coalescing into aggregates. Images taken from Movie EV2, which is representative of two additional Movies EV3 and 4. (B) Expanded image of the dashed box in the 1.5 min frame of (A). White arrows indicate substantial initial accumulation of fluorescent PC particles along filopodia extending from the cell bodies. (C) Expanded image of the dashed box in the 4.5 min frame of (A). White arrows indicate faint staining of filopodia by fluorescence PC particles relative to the accumulation of fluorescent PC particles in the cell bodies. (D) Fluorescent microscopy images after the addition of fluorescent saturated PC particles that do not induce aggregation. The red channel shows TdTomato-expressing *Capsaspora* cells. The white channel shows fluorescent saturated DPPC particles (DPPC/TopFluorPC [20:1 (w/w)]), which were not visible because these fluorescent PCs with saturated tail lipids are not taken up, correlating with lipids failing to induce aggregation. See Movie EV5.

(perhaps even unicellular microbes). Modest evidence from the literature supports this link between aggregation and feeding, as *Capsaspora* has been shown to form apparent aggregates onto schistosome prey, and the aggregates disperse after the prey has been consumed (Quick et al, 2025; Stibbs et al, 1979). This lipid-activated aggregation behavior may even have evolved in the ancestors of *Capsaspora* to efficiently feed on unicellular prey in a pre-metazoan world. To test this hypothesis, work is currently underway in our laboratory to discern if other filastereans also aggregate in response to zwitterionic lipids.

We note that signaling by pure lipid vesicles is an intriguing complement to the now-extensive field of signaling via extracellular vesicles (EVs) (Yanez-Mo et al, 2015). EVs are believed to be delivery vehicles for many signaling molecules, from lipids to carbohydrates, proteins, and nucleic acids (Yanez-Mo et al, 2015). Generally, the signal is only a small fraction of the entire EV composition (Yanez-Mo et al, 2015). However, our work is a reminder that the common bulk structural lipids that compose the vesicle membranes may themselves function as potent signaling

cues. This behavior suggests a broader and potentially under-appreciated mechanism of intercellular communication that may extend well beyond *Capsaspora*.

Besides identifying the molecular inducers of aggregation, our findings revealed that *Capsaspora* initiates aggregation through a rapid, post-translational mechanism that induced cell–cell compaction, providing new insights into the early stages of *Capsaspora*'s aggregation process. The initial step of aggregation is clearly "activation" of proteins already present in the cells—not increased production of adhesion proteins and extracellular matrix material, as was previously suspected following gene expression analysis of mature aggregates (Bercedo-Saborido et al, 2025; Sebé-Pedrós et al, 2013; Suga et al, 2013). However, the previously reported changes in gene expression likely do arise as aggregates mature—both in *Capsaspora* and its filasterean relative *Ministeria vibrans* (Bercedo-Saborido et al, 2025; Li et al, 2025; Sebé-Pedrós et al, 2013). Therefore, these data suggest multiple stages of aggregate maturation (Fig. 6B). An initial cell–cell contraction event rapidly assembles cells in a tight colony within a few minutes. Subsequent

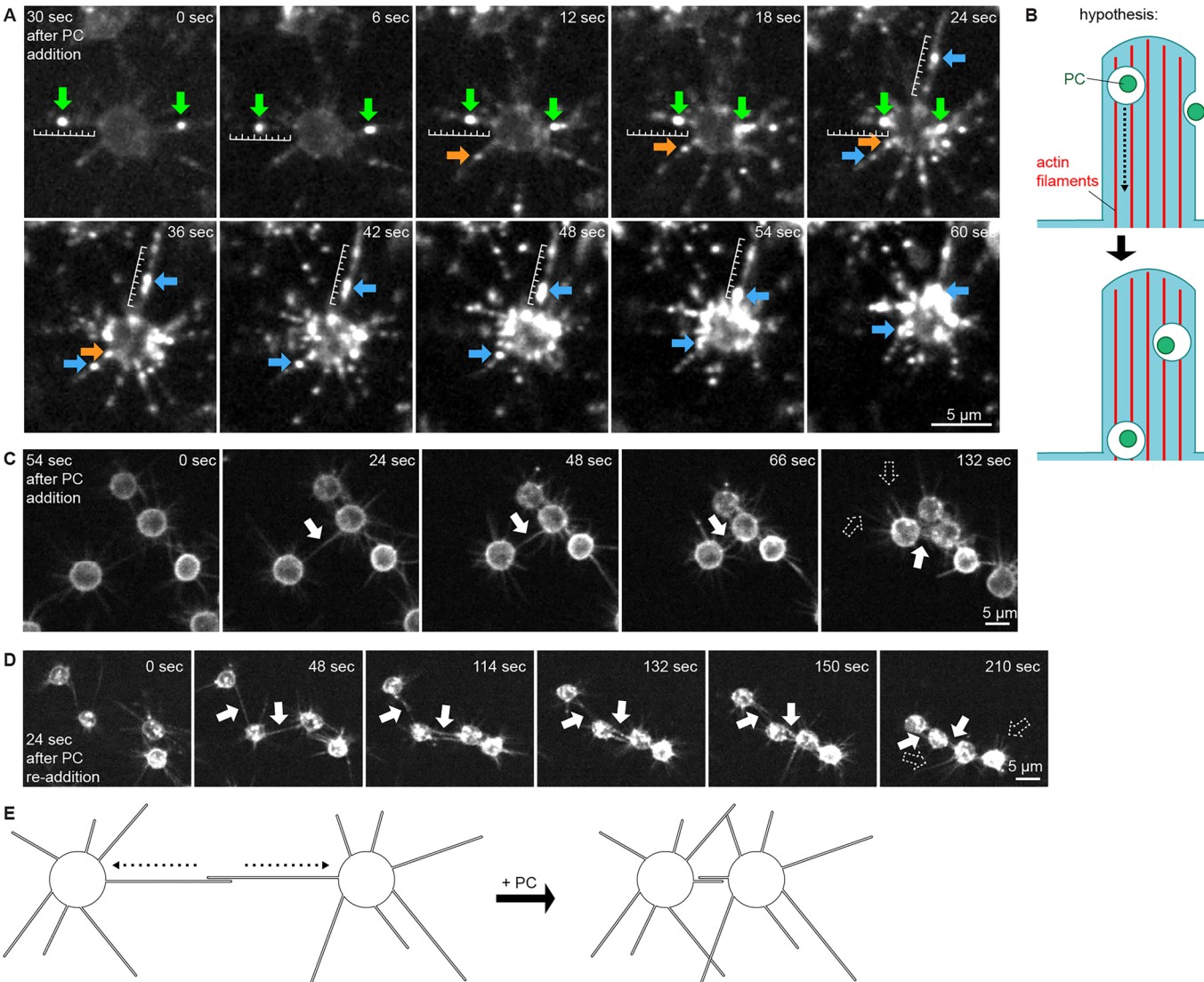

**Figure 5. Phosphatidylcholine is trafficked along filopodia and induces contacting filopodia to retract and pull adjacent cells together.**

(A) Time series of fluorescent microscopy images after addition of fluorescent PC particles (DOPC/TopFluorPC [20:1 (w/w)]) to cells containing an NMM-mVenus membrane marker (Phillips and Pan, 2024). Individual PC puncta are labeled with colored arrows to track movement across panels. Small-scale bars (5 µm with major ticks every 1 µm) indicate the rate of movement of fluorescent PC puncta along filopodia. Images taken from Movie EV6, which is representative of three Movies EV7–9. (B) Cartoon illustrating hypothesized endocytosis and trafficking of PC vesicles in filopodia. (C, D) Time series of fluorescent microscopy images after addition of non-fluorescent DOPC particles to membrane-labeled NMM-mVenus *Capsaspora* cells (Phillips and Pan, 2024). Images taken from Movies EV12, 13, which are representative of three Movies EV14–16. (C) Shows cells that aggregated after an initial addition of 100 µg/mL POPC. (D) Shows a different set of cells that aggregated after a second addition of 100 µg/mL POPC. Solid white arrows indicate intercellular filopodia contacts that contract, pulling adjacent cells together. Dashed outlined white arrows indicate filopodia that do not contact other cells and remain extended. (E) Cartoon illustrating the retraction of contacting filopodia upon the addition of phosphatidylcholine vesicles.

gene expression changes further mature the colony, which may include further filopodial retraction and production of extracellular matrix, among other physiological changes (Bercedo-Saborido et al, 2025; Li et al, 2025; Sebé-Pedrós et al, 2013).

The quick filopodial retraction observed here suggests a possible role for protein phosphorylation to regulate aggregation. Previous work reported that genes involved in regulating the actin cytoskeleton have many phosphosites, suggesting a complex phosphoregulation of these genes (Sebé-Pedrós et al, 2016b). Their work further reported differential phosphorylation of the "actin

cytoskeleton reorganization" gene ontology term between aggregates and non-aggregated cells. Therefore, we hypothesize that a change in protein phosphorylation leads to the rapid retraction of actin-filled filopodia. This hypothesis requires more thorough testing under our rapid chemically induced aggregation conditions.

In line with the nutritional potential of lipids, we have found that *Capsaspora* endocytoses both LDLs and PC vesicles. The aggregation response appears dependent on the clathrin-mediated endocytosis of these lipids, perhaps indicating that a lipid-sensing receptor signals from within endocytic vesicles (Esch and Fröhlich,

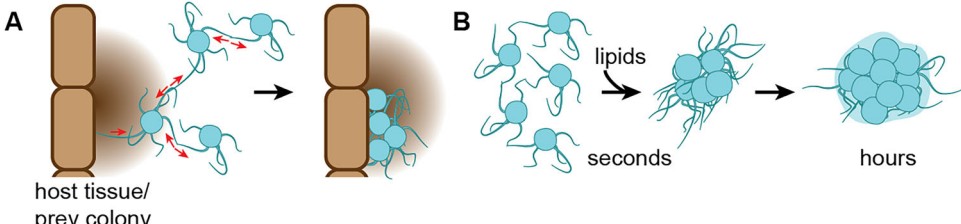

**Figure 6. Generality and kinetics of *Capsaspora* aggregation.**

(A) Immediate retraction of filopodia that connect neighboring cells leads to aggregation in response to zwitterionic lipids; this response may generally improve *Capsaspora* feeding efficiency by increasing cell density around any nutrient-rich host tissue or colony of prey cells. (B) *Capsaspora* aggregation likely involves various stages of development; initial aggregation of cells is followed by changes in gene expression that may include increased expression of cell adhesion machinery and extracellular matrix (Bercedo-Saborido et al, 2025; Li et al, 2025; Sebé-Pedrós et al, 2013; Suga et al, 2013).

2018). The convergence of lipid particle endocytosis and cellular signaling is reminiscent of lipoproteins and their receptors in animals (Lane-Donovan et al, 2014; May et al, 2005; Mineo, 2020). Several proteins related to the low-density lipoprotein receptor (LDLR) are responsible for lipoprotein uptake and/or cellular signaling processes (Nykjaer and Willnow, 2002). *Capsaspora* possesses transmembrane proteins that harbor the cytosolic NPXY motif, which classically drives endocytosis of LDLR homologs and some signaling processes (May et al, 2005). The large repetitive structure of lipid vesicles may also prove ideal for the multivalent interactions that assemble receptor multimers at the cell surface to activate endocytosis and/or signaling from LDLR homologs (May et al, 2005). Beyond LDLRs, signaling by many cell surface receptors (e.g., GPCRs and receptor tyrosine kinases) intersects with endocytosis in complex ways (Villasenor et al, 2016). Work is underway to identify potential receptor(s) in *Capsaspora*. Since the inducer is a lipid, it may alternatively incorporate into the *Capsaspora* plasma membrane, causing domains of different lipid composition that lead to membrane protein clustering and the initiation of a cellular aggregation signal. Future work will determine if PCs incorporate into the plasma membrane and/or endosomal membranes of *Capsaspora*.

Although much of the cellular process for aggregation remains a 'black box', our results indicate that filopodia rapidly retract to coalesce cells together. This finding is consistent with previous work by us and others that has shown connected and intertwined filopodia in *Capsaspora* and *M. vibrans* aggregates (Li et al, 2025; Phillips and Pan, 2024; Phillips et al, 2022; Ros-Rocher et al, 2021). From an evolutionary biology perspective, this is remarkable given that filopodia are essential for cell–cell adhesion in the developmental processes of diverse animals (Brunet and Booth, 2023). In fact, filopodia were initially observed in sea urchin gastrulation (Jacinto and Wolpert, 2001), where they play both cell–cell signaling roles and serve to pull the endoderm to the ectoderm (Hardin and McClay, 1990). Filopodial cell–cell contacts have later proven essential to draw cells together in the morphogenesis and wound-healing of cnidarians (Magie et al, 2007), Drosophila (Wood et al, 2002), nematodes (Raich et al, 1999), and even mammals (Vasioukhin et al, 2000). As we observed with *Capsaspora*, contact with a neighboring cell can trigger filopodial retraction in some animal cells (Bornschlogl et al, 2013). It remains to be seen whether *Capsaspora* filopodia retract by actomyosin contraction and/or actin treadmilling (or another mechanism).

Work by Phillips and Pan showed that aggregates loosen upon the addition of blebbistatin (Phillips and Pan, 2024), which lends some support to the actomyosin contraction hypothesis. We also do not know which proteins mediate cell–cell contacts between filopodia, nor do we know how these regulate filopodial retraction in conjunction with lipid signals. Addressing these mechanistic questions may shed light on the evolutionary origins of filopodia-based cell–cell adhesion, offering insight into the pre-metazoan innovations that may have preceded multicellularity in animals.

## Conclusion

In sum, we discovered that *Capsaspora* aggregation is induced by a wide panel of abundant zwitterionic diacyl phospholipids. The diverse nature of the inducing lipids suggests that the aggregation response might be general to many nutrient-rich environments (not just host snails) and may enable *Capsaspora* (or its ancestors) to efficiently prey on other organisms. Moreover, *Capsaspora* lipid-induced aggregation utilizes clathrin-mediated endocytosis of lipids. This aggregative process occurs remarkably fast (within seconds) via post-translational processes that appear to include retraction by filopodia bound to neighboring cells. The sudden cell–cell compaction observed in early aggregation requires further analysis for comparison with similar cell–cell adhesion and contraction behaviors in multicellular animals. Overall, these insights advance our understanding of *Capsaspora* cellular aggregation, which is a potential key to the unicellular-to-multicellular transition in the animal lineage and may also be an essential behavior to support an anti-schistosome symbiosis.

## Methods

**Reagent and tools table**

| Reagent/resource | Reference or source | Identifier or catalog number |
|---|---|---|
| **Experimental models** | | |
| *Capsaspora owczarzaki* | ATCC | 30864 |
| *Capsaspora owczarzaki* NMM-mVenus | Jonathan Phillips & Duojia Pan[35] | |
| *Capsaspora owczarzaki* TdTomato | Our laboratory[12] | |

| Reagent/resource | Reference or source | Identifier or catalog number |
|---|---|---|
| **Chemicals, enzymes, and other reagents** | | |
| 1-palmitoyl-2-oleoyl-glycero-3-phosphocholine [POPC] | Avanti Polar Lipids | 850457C |
| 1,2-dioleoyl-sn-glycero-3-phosphocholine [DOPC] | Avanti Polar Lipids | 850375P |
| 3-palmitoyl-2-oleoyl-sn-glycero-1-phosphocholine [Ent-POPC] | Avanti Polar Lipids | 850855C |
| 1,2-di-O-(9Z-octadecenyl)-sn-glycero-3-phosphocholine [diether-PC] | Avanti Polar Lipids | 999989P |
| 2-((2,3-bis(oleoyloxy)propyl) dimethylammonio)ethyl ethyl phosphate [DOCPe] | Avanti Polar Lipids | 850312P |
| 1-palmitoyl-2-oleoyl-sn-glycero-3-ethylphosphocholine (chloride salt) [EPC] | Avanti Polar Lipids | 890705P |
| N-oleoyl-D-erythro-sphingosylphosphorylcholine [18:1 SM] | Avanti Polar Lipids | 860587C |
| N-palmitoyl-D-erythro-sphingosylphosphorylcholine [16:0 SM] | Avanti Polar Lipids | 860584P |
| 1-palmitoyl-2-oleoyl-sn-glycero-3-phosphate (sodium salt) [POPA] | Avanti Polar Lipids | 840857C |
| 1-palmitoyl-2-oleoyl-sn-glycero-3-phosphoethanolamine [POPE] | Avanti Polar Lipids | 850757P |
| Phosphocholine chloride calcium salt tetrahydrate | Sigma Aldrich | P0378 |
| Choline chloride | Fisher Scientific | A1582822 |
| 1-palmitoyl-2-hydroxy-sn-glycero-3-phosphocholine [16:0 LPC] | Avanti Polar Lipids | 855675P |
| 1-oleoyl-2-hydroxy-sn-glycero-3-phosphocholine [18:1 LPC] | Avanti Polar Lipids | 845875P |
| 1,2-dimyristoleoyl-sn-glycero-3-phosphocholine [14:1 PC] | Avanti Polar Lipids | 850346C |
| 1,2-dipalmitoleoyl-sn-glycero-3-phosphocholine [16:1 PC] | Avanti Polar Lipids | 850358C |
| 1,2-dilinolenoyl-sn-glycero-3-phosphocholine [18:3 PC] | Avanti Polar Lipids | 850395C |
| 1,2-dieicosenoyl-sn-glycero-3-phosphocholine [20:1 PC] | Avanti Polar Lipids | 850396C |
| 1,2-diarachidonoyl-sn-glycero-3-phosphocholine [20:4 PC] | Avanti Polar Lipids | 850397C |
| 1,2-dimyristoyl-sn-glycero-3-phosphocholine [DMPC] | Avanti Polar Lipids | 850345P |
| 1,2-dipalmitoyl-sn-glycero-3-phosphocholine [DPPC] | Avanti Polar Lipids | 850355C |
| 1,2-distearoyl-sn-glycero-3-phosphocholine [18:0 PC] | Avanti Polar Lipids | 850365C |
| 1,2-di[(8Z)octadecenoyl]-sn-glycero-3-phosphocholine [18:1 (8cis) PC] | Avanti Polar Lipids | 792486C |
| 1,2-dielaidoyl-sn-glycero-3-phosphocholine [18:1 (9trans) PC] | Avanti Polar Lipids | 850376C |
| 1,2-divaccenoyl -sn-glycero-3-phosphocholine [18:1 (11cis) PC] | Avanti Polar Lipids | 790626C |
| 1,2-dinervonoyl-sn-glycero-3-phosphocholine [24:1 PC] | Avanti Polar Lipids | 850399C |

| Reagent/resource | Reference or source | Identifier or catalog number |
|---|---|---|
| Dynole 34-2 | Neta Scientific | CAYM-34073-1 |
| Pitstop II | Abcam | ab120687 |
| Dynasore | Neta Scientific | CAYM-14062-5 |
| MiTMAB | Abcam | ab120468 |
| OctMAB | Abcam | ab120468 |
| CK-666 | Sigma | 182515-25MG |
| Chlorpromazine | Sigma | C8138-5G |
| Ponesimod | Sigma | ADV638391874-25MG |
| ABMA | Med Chem Express | HY-124801 |
| EGA | Sigma | 5093060001 |
| Monensin | Sigma | M5273-500MG |
| Cycloheximide | Sigma | 01810-1G |
| pHrodo red LDL | Fisher Scientific | I34360 |
| Hoechst DNA stain | Fisher Scientific | BDB561908 |
| TopFluor-PC | Avanti Polar Lipids | 810281 |
| BODIPY LDL | Fisher Scientific | L3483 |
| **Software** | | |
| Fiji Imaging Software | | version 2.1.0/1.53c |
| Graphpad Prism | | version 10.6.1 |
| Ffmpeg tool | | build 20200213-6d37ca8 |

## Cell strain and growth conditions

*Capsaspora owczarzaki* cell cultures (strain ATCC®30864) were grown axenically in 25-cm² culture flasks with 6 mL ATCC media 1034 (modified PYNFH medium: 10 g/L peptone, 10 g/L yeast extract, 1 g/L yeast ribonucleic acid, 15 mg/L folic acid, 1 µg/mL hemin, 2.66 mM $KH_2PO_4$, and 3.52 mM $Na_2HPO_4$ at pH 6.5 in water) containing 10% (v/v) heat-inactivated fetal bovine serum (FBS), hereafter *growth media*, in a 23 °C incubator. Adherent-stage cells (filopodiated amoebae) at the exponential growth phase were obtained by passaging ~100–150 µL of adherent cells at ~90% confluence in 6 mL of growth media and grown for 24–48 h at 23 °C until ~100% confluent.

## General aggregation assay methods

All aggregation assays were performed at room temperature. Brightfield imaging was performed using the following instruments: Leica DMi1 inverted microscope with an MC120 HD camera, Leica DMIL inverted microscope with Flexacam C3 camera, an Olympus OSR spinning disk confocal microscope with a Hamamatsu Flash 4 V2 camera, an Applied Precision DeltaVision OMX Super Resolution 3D-SIM system with a pco.edge 4.2 camera, an Incucyte S3 Live-Cell Analysis System, and an A1 Nikon Scanning Confocal with a Hammamatsu Orca-Flash 4.0 sCMOS camera. Depending on well size and microscope used, each well was imaged at up to

three distinct locations using ×5 or ×10 magnification. Average aggregate areas were typically measured by batch processing with a standard macro script in Fiji Imaging Software (Ros-Rocher et al, 2021; Schindelin et al, 2012) (see "Image analysis" below).

## Aggregation assay on ultra-low attachment plates

Two days before the assay, 100% confluent adherent cells growing in 25-cm² culture flasks were given fresh growth media (termed the "feed step"). One day before the assay, cells were washed and resuspended in FBS-free assay media and allowed to sit overnight (termed the "starve step"). After starvation, the day of the assay, $8 \times 10^5$ cells were seeded in 180 µL of FBS-free media per well in a 96-well ultra-low attachment microplate (#CLS3474, Corning) and allowed to settle for 2 h. Putative aggregation inducers were added such that the total volume in a well was 200 µL. Typically, aggregates were assessed by microscopy after 90 min.

## Image analysis for aggregation assays

Average aggregate areas were typically measured by batch processing with a standard macro script in Fiji Imaging Software version 2.1.0/1.53c (Ros-Rocher et al, 2021; Schindelin et al, 2012). Briefly, the macro steps included: set the scale of the image appropriate for the microscope conditions, convert the image to binary, analyze particles (size 0-infinity), and export results to the clipboard. Example images from the processing steps and frequency distributions of resulting "aggregate" areas are presented in Fig. EV1. A copy of the FIJI macro is available upon request.

## Structure–activity relationship studies (related to Figs. 1, EV2, and EV3)

### Preparing lipid vesicles from pure lipid samples (related to Figs. 1, EV2, and EV3)

Lipids commercially obtained (pure synthetic lipids) were dissolved in chloroform in glass vials. 0.2 mg was transferred to a 1.7 mL microcentrifuge tube or a 1-dram glass vial. The chloroform was evaporated using a gentle stream of nitrogen, and the resulting lipid film was then resuspended in 1 mL each of 1× PBS or FBS-free assay media. The tubes were vortexed for 30 s before sonication. Lipid solutions were sonicated on ice with a 50% duty cycle at medium setting for 10 min using a single probe attached to a Branson Sonifier Cell-Disruptor 185 or in sets of 4 using a 4-probe sonicator horn attached to a Fisherbrand Model 505 Sonic Dismembrator (FB505A110). During sonication, tubes were kept on ice, and tube bottoms were set about 3 mm from the sonicator tip. Lipid solutions went from slightly cloudy to clear after sonication. Sonicated lipids were stored at 4 °C for no more than 2 days before use.

Crude lipids were concentrated 10× using a > 30 kDa cutoff filter immediately before addition into aggregation assay wells. The concentration of lipids was reported in µg/mL based on the mass of lipid/volume of the assay well.

### Testing lipid vesicles for aggregation (related to Figs. 1, EV2 and EV3)

Sonicated lipids were concentrated 10× using an Amicon Ultra 30 kDa cutoff filter (Millipore UFC5030) and diluted with 1× PBS

to the desired concentration before testing. The standard aggregation assay on ultra-low attachment plates was used to test for aggregation induction. Aggregates were induced using the desired lipid concentration (calculated in µg/mL of lipid) and 5% (v/v) of 1× PBS or sonicated FBS-free assay media was used as a negative control, depending on what the lipids were prepared in. Aggregates in triplicate wells were imaged every 30 min, and analysis was performed on images from T-90 min Average aggregate areas were measured using a macro in FIJI as described above.

### Transmission electron microscope (TEM) imaging of prepared lipid vesicles (related to Fig. EV2)

Before preparation of samples, Formvar/Carbon film 300 mesh Nickel Grids (Electron Microscopy Sciences FCF300-Ni-25) were ionized using a Pelco easiGlow Discharge System at 0.3 mbar and 15 mAmps for 2 min. In total, 10 µL of prepared lipid samples (sonicated for 10 min) were added to the ionized grids and allowed to sit for 5 min. Excess sample was removed from the grids by gently touching the Whatman filter paper (Sigma, WHA1001329) to the edge. Immediately after removing samples (before grid dries) 10 µL of 2% (v/v) Uranyl Acetate pre-diluted in water was added to the grids and allowed to sit for 10 s. Then the stain was removed with the edge of the Whatman filter paper. Grids were allowed to dry completely before imaging (about 10 min). Prepared grids were imaged immediately after preparation using a JEOL-JEM 1010 transmission electron microscope (TEM) with an 80 kV operating voltage, equipped with a 1k x 1k Gatan CCD camera (MegaScan model 794) using a tungsten filament as its electron source.

## Post-translational regulation studies (related to Figs. 2, EV4; Movie EV1)

### Videos of Capsaspora aggregation (related to Fig. 2A; Movie EV1)

*Capsaspora* cells were prepared for a standard aggregation assay as described above. Cells were allowed to settle in ultra-low attachment plates for 2 h. Plates were placed on a Leica DMi1 inverted microscope and focused on ×10 magnification with brightfield illumination. Video recording was started, and 5% (vol/vol) FBS was added gently to the edge of the assay well about 10 s after recording had started, and cells were recorded for 5 min continuously. Video was converted to black and white to show cells more clearly and compressed from its original size before upload using the ffmpeg tool build 20200213-6d37ca8 (FFmpeg Developers, 2007).

### Inhibition of new protein synthesis using cycloheximide (related to Figs. 2B and EV4)

Normal aggregation assays were set up according to the protocol above. A dilution series of the protein synthesis inhibitor cycloheximide (Sigma 01810) was created in water. After allowing cells to settle in assay plates for 2 h, cells were treated with cycloheximide inhibitors for 30 min before inducing aggregation with 5% (v/v) FBS. Images for aggregation assay were assessed 90 min after the addition of FBS.

### Measuring new protein synthesis using incorporation of radioactive amino acid [³⁵S] methionine (related to Fig. EV4)

Cells were prepared as if for a standard aggregation assay as described above. About 480 million cells were pelleted with

centrifugation at 1000×*g* and resuspended into 12 mL of FBS-free assay media. These cells were divided into 24 portions, each with 0.5 mL of cells (about 20 million cells) in 1.7-mL centrifuge tubes. Cycloheximide dilutions were prepared in water and added to cells to reach the desired final concentration. A negative control with water and a positive control with 4% formaldehyde were also prepared. Cells were treated with cycloheximide (or formaldehyde) for 30 min. Then, 20.4 µCi of [$^{35}$S] L-methionine (Revvity/Perkin Elmer NEG009A500UC) was added to each tube and allowed to incubate for 2 h. Cells were then pelleted by centrifugation and resuspended in 0.5 mL of cold cell lysis buffer (NP-40 buffer + 1 mg/mL SDS + 5 mg/mL sodium deoxycholic acid + 0.38 mg/mL EGTA). Cells were incubated on ice for 30 min and then pipetted vigorously before centrifugation at 15,000×*g* for 10 min to remove debris. The supernatant was then mixed 1:1 with TCA (acetone + 20% trichloroacetic acid + 0.07% 2-mercaptoethanol) and allowed to sit for 45 min on ice to precipitate proteins. After precipitation, samples were centrifuged at 15,000×*g* for 10 min and the pellet was washed with 1 mL of acetone before air drying. The dry pellets were then resuspended in 40 µL of PBS buffer, and the whole sample was spotted onto Whatman filter paper. The paper was allowed to dry completely before covering with plastic wrap and placing in a phosphor plate chamber overnight. The next day, the phosphor plate was imaged using a Typhoon 9210 Variable Mode Imaging System. Quantification of the intensity of the dot blot was performed using Fiji image software by measuring the average intensity of each spot.

## Endocytosis studies (related to Figs. 3 and EV5)

### Uptake of BODIPY-labeled fluorescent LDL (related to Fig. 3B)

Cells were prepared as if for a normal aggregation assay as described above. Cells were seeded in assay media in chambered glass microscope slides (Ibidi 80827) and allowed to settle for 2 h. Assay media was replaced with Chernin's balanced salt solution (CBSS + , 2.8 g/L sodium chloride, 0.15 g/L potassium chloride, 0.07 g/L sodium phosphate dibasic, 0.45 g/L magnesium sulfate heptahydrate, 0.53 g/L calcium chloride dihydrate, 0.05 g/L sodium bicarbonate, 1 g/L glucose, and 1 g/L trehalose at pH 7.2 in water) (Chernin, 1963) buffer for imaging by sequential aspiration and replacement until liquid in wells was clear (about 3 times) and the total volume in the well was 400 µL. Storage buffer for fluorescent low-density lipoprotein (BODIPY LDL, Fisher L3483) was replaced with CBSS+ buffer before adding. 80 µg/mL of BODIPY LDL was added to the microscope slide and allowed to incubate for 30 min, then 5 µg/mL of Hoechst DNA stain (Fisher BDB561908) was added and allowed to sit for 5 min before removing and replacing with fresh buffer immediately before imaging on an OMX-SR 3D-SIM system with an excitation laser of 488 nm for BODIPY LDL and 405 nm for Hoechst at a magnification of ×60. Images are example slides with a scale bar of 5 µm.

### Uptake of fluorescent PC particles (related to Fig. 3C)

Cells were prepared as if for a normal aggregation assay as described above. Fluorescent TopFluor-PC (Avanti 810281) was doped in at a ratio of 20 POPC to 1 TopFluor-PC before making vesicles via sonication as described above. Cells were seeded in FBS-free assay media into chambered #1.5H microscope slides (Ibidi 80827) and allowed to settle for 2 h. Assay media was replaced with Chernin's balanced salt solution (CBSS + ) buffer for imaging by sequential aspiration and replacement

until liquid in wells was clear (about 3 times) and the total volume in the well was 400 µL. 100 µg/mL of TopFluor-PC:POPC particles were added to cells and allowed to incubate for 30 min before imaging on an OMX-SR 3D SIM super resolution microscope system with a 488 nm excitation laser for TopFluo-PC and a 405 nm laser for Hoechst with a magnification of ×60. Images shown are example wells with a scale bar of 5 µm.

### Inhibition of uptake of pHrodo LDL particles using Dynole 34-2 and monensin (related to Fig. 3D,E and EV5O)

Cells were prepared for a normal aggregation assay as described above. Cells were seeded in FBS-free assay media into ultra-low attachment assay plates and allowed to settle for 2 h. A dilution series of Dynole 34-2 (Neta Scientific, CAYM-34073-1) in DMSO was prepared and added to assay wells in triplicate. In all cases, DMSO concentrations were kept constant at 0.5% in assay wells except for one control with no DMSO and one control with no pHrodo LDL. Alternatively, cells were treated with 10 µM monensin (Sigma, M5273) in a final concentration of 1% DMSO. Cells were incubated with inhibitors for 30 min. The storage buffer for pHrodo LDL (Fisher, I34360) was replaced with PBS buffer immediately before using in the assay. 80 µg/mL of pHrodo LDL was added to each well and allowed to sit for 30 min. Cells were then fixed using 4% formaldehyde, and assay media were replaced with CBSS+ for imaging. In total, 5 µg/mL of Hoechst DNA stain (Fisher, BDB561908) was added and allowed to sit for 5 min before removing and replacing with fresh CBSS+ buffer. Cells were imaged on an Olympus microscope with widefield CFP and RFP lamps at ×20 magnification. Images shown are example images of cells in wells with DMSO but no Dynole 34-2. Images were quantified in batch using Fiji Imaging Software version 2.1.0/1.53c. Briefly, the macro was set to find edges, convert to binary, despeckle, and count particles in the Hoechst channel and in the pHrodo channel. Plot is the percent of Hoechst-stained cells that were also stained with pHrodo; the final result was analysis of duplicate images of triplicate wells (*n* = 6).

### Inhibition of Capsaspora aggregation using chemical inhibitors of endocytosis (related to Figs. 3F,G and EV5)

Normal aggregation assays were set up according to the protocol above. Inhibitors were purchased and prepared according to solubility. In cases where inhibitors were soluble in DMSO, the final DMSO concentration in assay wells was kept at a constant 0.5% and a DMSO control was included. In cases where inhibitors were soluble in ethanol, the final concentration of ethanol in assay wells was kept at a constant 1% and an ethanol control was included. After allowing cells to settle in assay plates, cells were treated with chosen inhibitors for 30 min before inducing aggregation with 5% (v/v) FBS. For Fig. EV5L–N, aggregation was instead induced with 100 µg/mL POPC vesicles (prepared via sonication, as above). Assays were performed in triplicate and aggregates were analyzed 90 min after induction.

## Vesicle trafficking and filopodia retraction studies (related to Figs. 4 and 5; Movies EV2–16)

### Timelapse imaging of Capsaspora uptake of fluorescent PC particles into TdTomato-labeled cells (related to Fig. 4; Movies EV2–5)

*Capsaspora* cells constitutively expressing TdTomato fluorescent protein (Kidner et al, 2024) were prepared as if for a standard aggregation assay as described above. About $1.6 \times 10^6$ cells were

added in FBS-free assay media to a #1.5H clear chambered glass microscope slide (ibidi 80827) and allowed to settle for 2 h. Directly before imaging, assay media were removed and replaced with CBSS+ buffer to reduce background fluorescence. Fluorescent PC particles were made by combining either DOPC (Avanti Polar Lipids 850375 P) or DPPC (Avanti Polar Lipids 850355 C) with TopFluorPC (Avanti Polar Lipids 810281 C) in a ratio of 20 DOPC/DPPC to 1 TopFluorPC (w/w). Lipid vesicles were prepared by mixing 2 mg of total lipids dissolved in chloroform in a glass vial, evaporating the chloroform with a gentle stream of nitrogen to form a thin film, and then resuspending lipids in 1 mL of PBS buffer. Lipids were then extruded through 100 nm pores in an extruder (Avanti Polar Lipids 610000). Lipids were used in the assay on the same day. Cells were placed on an Olympus spinning disk microscope and images were set up to be taken of each channel (561 nm excitation for TdTomato and 488 nm excitation for TopFluorPC) and imaging was started. After 2 imaging cycles (~12 s) fluorescent PCs (100 µg/mL final concentration) were added gently to one corner of the well while the microscope recording was still running. Images were taken every 5.6 s for ~15 min. Stacks of images were compressed and converted into videos using Fiji software.

### Timelapse imaging of Capsaspora uptake of fluorescent PC particles into NMM-mVenus-labeled cells (related to Fig. 5A; Movies EV6–9)

*Capsaspora* cells constitutively expressing NMM-mVenus fluorescent fusion (Phillips and Pan, 2024) were prepared as if for a standard aggregation assay as described above. About $1.6 \times 10^6$ cells were added in FBS-free assay media to a #1.5H chambered glass microscope slide (ibidi 80827) and allowed to settle for 2 h. Directly before imaging, assay media were removed and replaced with CBSS+ buffer to reduce background fluorescence. Fluorescent PC particles were made by combining DOPC (Avanti Polar Lipids 850375 P) with TopFluorPC (Avanti Polar Lipids 810281 C) in a ratio of 20 DOPC/DPPC to 1 TopFluorPC (w/w). Lipid vesicles were prepared by mixing 2 mg of total lipids dissolved in chloroform in a glass vial, evaporating the chloroform with a gentle stream of nitrogen to form a thin film, and then resuspending lipids in 1 mL of PBS buffer. The tubes were vortexed for 30 s before sonication. Lipid solutions were sonicated on ice with a 50% duty cycle at medium setting for 10 min using a single probe attached to a Branson Sonifier Cell-Disruptor 185. During sonication, tubes were kept on ice and tube bottoms were set about 3 mm from the sonicator tip. Lipid solutions went from slightly cloudy to clear after sonication. Lipids were used in the assay ~4 days after sonication (stored at 4 °C). Cells were placed on an Olympus spinning disk microscope, and images were set up to be taken of the green channel (488 nm excitation) and imaging was started. After 5 imaging cycles (~30 s) fluorescent PCs (100 µg/mL final concentration) were added gently to the center of the well while the microscope recording was still running. Images were taken every 5.9 s for ~6 min. Images were converted into videos using Fiji software.

### Timelapse imaging of Capsaspora failure to endocytose fluorescent PC particles upon treatment with an endocytosis inhibitor (related to Movies EV10 and 11)

*Capsaspora* cells (without NMM-mVenus) were prepared as if for a standard aggregation assay as described above. About $1.6 \times 10^6$ cells

were added in FBS-free assay media to a #1.5H chambered glass microscope slide (ibidi 80827) and allowed to settle for 2 h. Directly before imaging, assay media were removed and replaced with CBSS+ buffer to reduce background fluorescence. The CBSS+ buffer contained either 1% DMSO negative control or 1% of a 10 mM DMSO solution of Dynole 34-2, for a final concentration of 100 µM. Cells were incubated in this solution for 30 min before imaging. Fluorescent PC particles were made by combining POPC (Avanti Polar Lipids 850457 C) with TopFluorPC (Avanti Polar Lipids 810281 C) in a ratio of 20 POPC to 1 TopFluorPC (w/w). Lipid vesicles were prepared by mixing 2 mg of total lipids dissolved in chloroform in a glass vial, evaporating the chloroform with a gentle stream of nitrogen to form a thin film, and then resuspending lipids in 1 mL of PBS buffer. The tubes were vortexed for 30 s before sonication. Lipid solutions were sonicated on ice with a 50% duty cycle at medium setting for 10 min using a single probe attached to a Branson Sonifier Cell-Disruptor 185. During sonication, tubes were kept on ice, and tube bottoms were set about 3 mm from the sonicator tip. Lipid solutions went from slightly cloudy to clear after sonication. Lipids were used in the assay ~4 days after sonication (stored at 4 °C). Cells were placed on a Leica SP8 scanning confocal microscope at ×20 magnification and 6× zoom. Images were set up to be taken with the BODIPY fluorophore settings, and imaging was started. Fluorescent PCs (100 µg/mL final concentration) were added gently to the center of the well while the microscope recording was still running. Images were taken every 10 s for 61 images (10 min). Images were converted into videos using Fiji software.

### Timelapse imaging of Capsaspora aggregation through filopodia using NMM-mVenus-labeled cells (related to Fig. 5C,D; Movies EV12–16)

*Capsaspora* cells constitutively expressing NMM-mVenus fluorescent fusion (Phillips and Pan, 2024) were prepared as if for a standard aggregation assay as described above. About $1.6 \times 10^6$ cells were added in FBS-free assay media to a #1.5H chambered glass microscope slide (ibidi 80827) and allowed to settle for 2 h. Directly before imaging, assay media were removed and replaced with CBSS+ buffer to reduce background fluorescence. Non-fluorescent DOPC particles were made by adding 2 mg of DOPC (Avanti Polar Lipids 850375 P) dissolved in chloroform in a glass vial, evaporating the chloroform with a gentle stream of nitrogen to form a thin film, and then resuspending lipids in 1 mL of PBS buffer. The tubes were vortexed for 30 s before sonication. Lipid solutions were sonicated on ice with a 50% duty cycle at medium setting for 10 min using a single probe attached to a Branson Sonifier Cell-Disruptor 185 or in sets of 4 using a 4 probe sonicator horn attached to a Fisherbrand Model 505 Sonic Dismembrator (FB505A110). During sonication, tubes were kept on ice and tube bottoms were set about 3 mm from the sonicator tip. Lipid solutions went from slightly cloudy to clear after sonication. Lipids were used in the assay within a day of sonication. Cells were placed on an Olympus spinning disk microscope, and images were set up to be taken with the green channel (488 nm excitation), and imaging was started. After 14 imaging cycles (~84 s) PCs (100 µg/mL final concentration) were added gently to the center of the well while the microscope recording was still running. Images were taken every 5.9 s for ~30 min. Stacks of images were compressed and converted into videos using Fiji software.

## Data availability

All data are available within the article and source image data are collected in The BioImage Archive (Hartley et al, 2022): https://www.ebi.ac.uk/biostudies/bioimages/studies/S-BIAD2733.

The source data of this paper are collected in the following database record: biostudies:S-SCDT-10_1038-S44319-026-00760-1.

## Peer review information

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

## Acknowledgements

We thank the Light Microscopy Center at Indiana University for support in image acquisition and analysis (funding provided by the NIH grant NIH1S10OD024988-01). We also thank the Indiana University Nanoscale Characterization Facility, Electron Microscopy Center, Laboratory for Biological Mass Spectrometry, and Physical Biochemistry Instrumentation Facility for use of their instruments. We also thank the entire Gerdt lab for insights and support that helped advance this project. This work was supported by the National Institutes of Health (R35GM138376) to JPG. RQK was supported by an NIH training grant (T32GM131994). The content of this paper is solely the responsibility of the authors and does not necessarily represent the official views of the National Institutes of Health. NRR was supported by the European Union's Horizon Europe research and innovation funding program under a Marie Skłodowska-Curie Actions grant (FlexAggon, grant agreement ID: 101106415), by the Institut Pasteur, and by the CNRS (UMR3691).

## Author contributions

**Ria Q Kidner**: Conceptualization; Investigation; Visualization; Methodology; Writing—original draft; Writing—review and editing. **Eleanor B Goldstone**: Investigation. **Henry J Rodefeld**: Investigation. **Lorin P Brokaw**: Investigation. **Aria M Gonzalez**: Investigation. **Lalitha Sastry**: Investigation. **Ranojoy Baisya**: Investigation. **Núria Ros-Rocher**: Investigation; Writing—review and editing. **Joseph P Gerdt**: Conceptualization; Supervision; Funding acquisition; Visualization; Methodology; Writing—original draft; Writing—review and editing.

Source data underlying figure panels in this paper may have individual authorship assigned. Where available, figure panel/source data authorship is listed in the following database record: biostudies:S-SCDT-10_1038-S44319-026-00760-1.

## Disclosure and competing interests statement

The authors declare no competing interests.

# Expanded View Figures

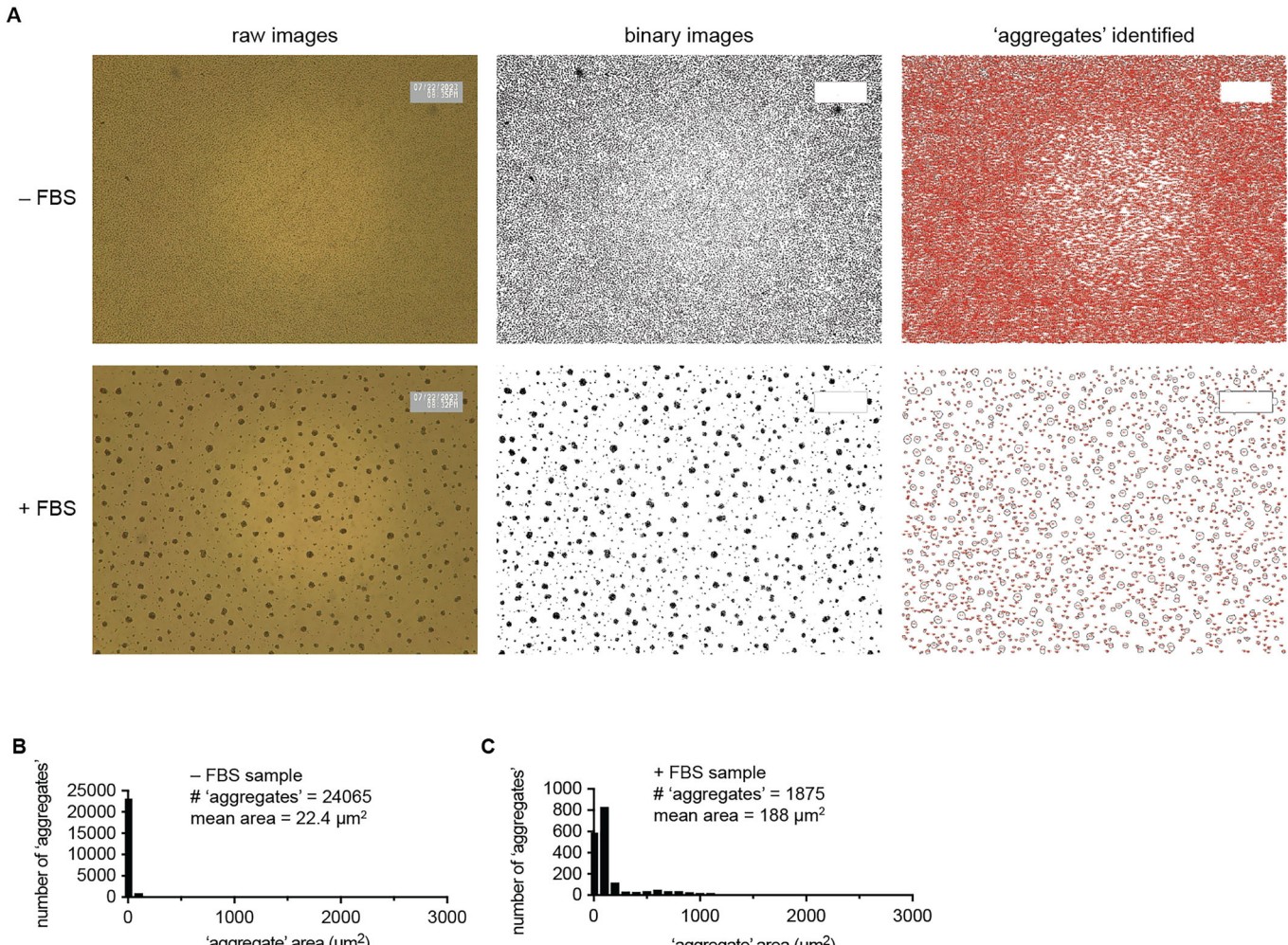

**Figure EV1. Image processing illustration.**

(A) Image analysis steps for uninduced cells (−FBS) and aggregation-induced cells (+ FBS), showing the raw images, images after binary processing, and finally the images after outlines were drawn around individual particles, which are each considered an "aggregate". (B) Histogram reporting the frequency distribution of "aggregates" (i.e., particles) in the −FBS sample. Nearly all "aggregates" are single cells with areas <50 μm². (C) Histogram reporting the frequency distribution of "aggregates" (i.e., particles) in the +FBS sample. In this case, most were >50 μm², with many near 1000 μm². Even though many aggregates contain hundreds of cells are present, the "mean area" for the image is only ~200 μm² due to the greater number of single cells and tiny aggregates. Nonetheless, "average area" distinguishes these aggregated cells from the non-aggregated condition (which had an average area ~20 μm²).

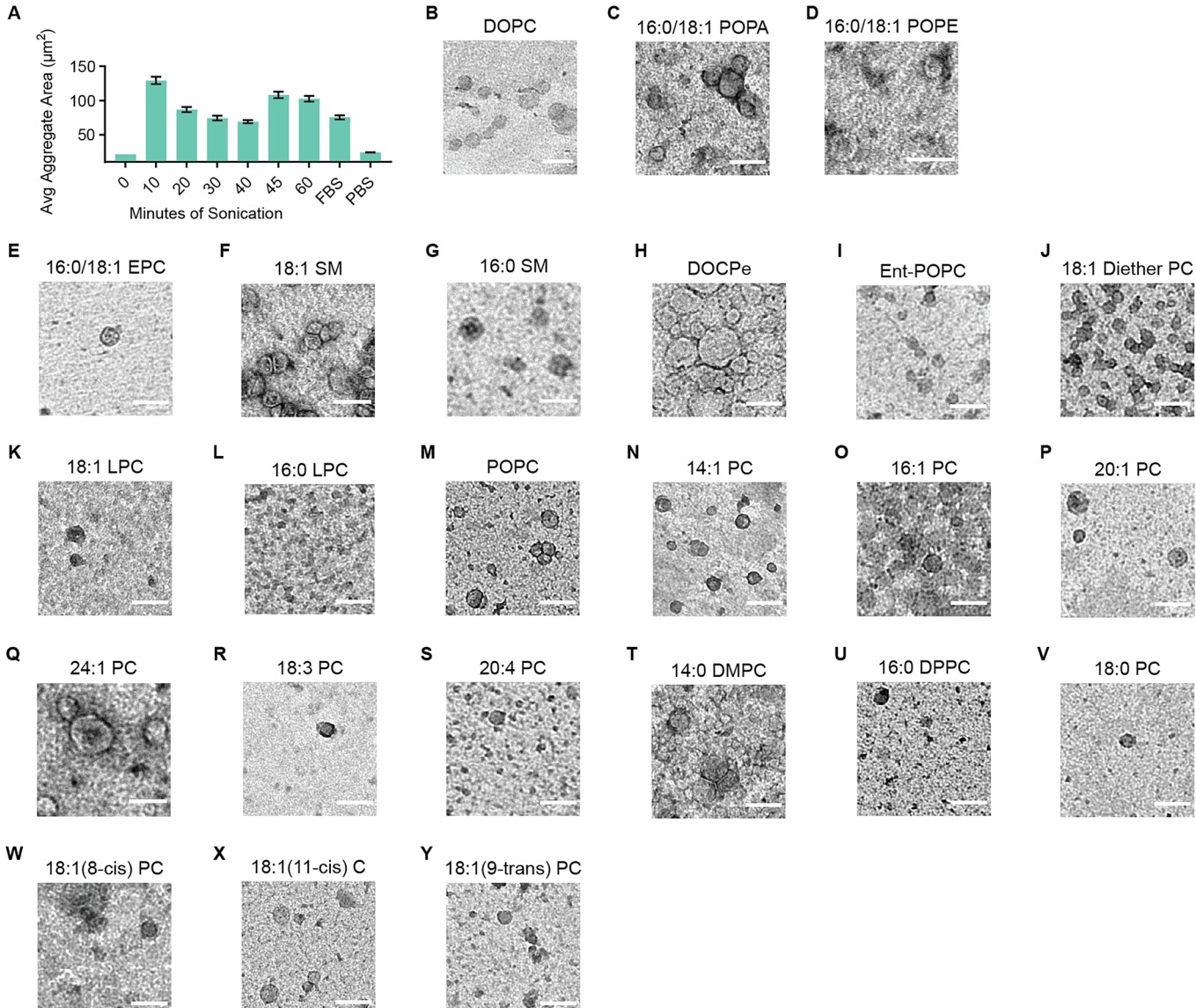

**Figure EV2. Sonication and TEM images for structure–activity relationship study.**

(A) Average aggregate area resulting from the addition of POPC prepared with varying sonication times. Preparation of PC lipids into vesicles by sonication is necessary for aggregation-induction; adding lipids without sonication failed to induce aggregation. 5% (v/v) FBS and 5% (v/v) PBS were used as positive and negative controls, respectively. Error bars represent SEM of biological triplicates ($n = 3$). For each timepoint, a single batch of sonicated vesicles was used. (B–Y) Transmission electron microscope (TEM) images of prepared lipid vesicles (sonicated for 10 min) stained with 1% uranyl acetate. Scale bars are 100 nm.

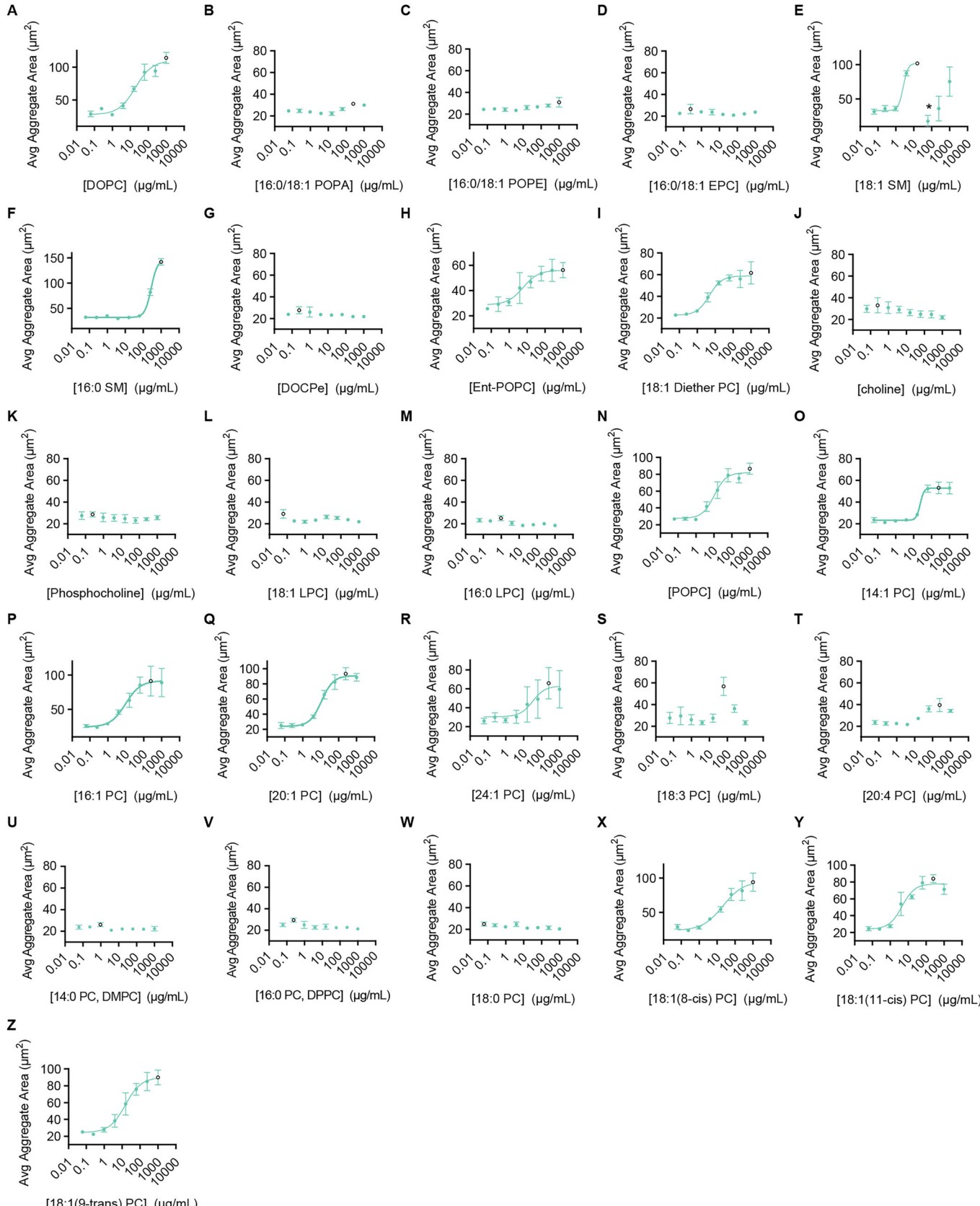

◀  **Figure EV3.  Individual aggregation curves for structure activity relationship study.**

(**A–Z**) Individual dose–response curves for all tested lipids (the white circle represents the largest aggregate value, which was the selected concentration reported in Fig. 1). Error bars represent SEM of biological triplicates ($n = 3$). For 18:1 SM in (**E**), high concentrations caused a single large aggregate of cells to form in some replicates. Since there was only a single aggregate, the "average" aggregate area was abnormally decreased by the individual cells in the well. This applies to concentrations greater than or equal to the point noted with an asterisk.

    

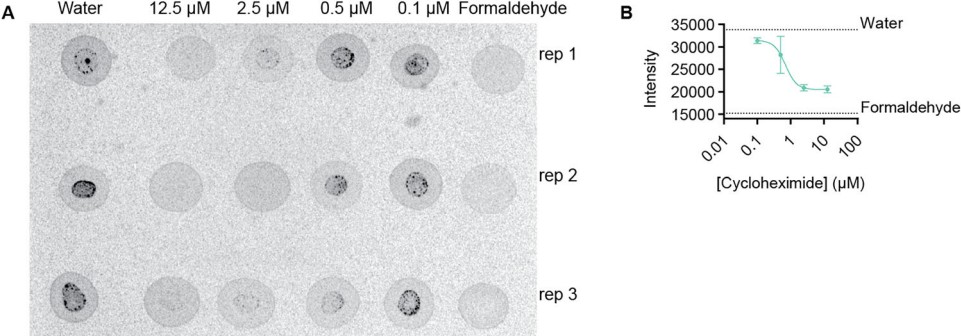

**Figure EV4. Cycloheximide inhibits protein synthesis in *Capsaspora*.**

(A) Phosphor plate image of incorporation of radioactive [$^{35}$S] methionine in cycloheximide-treated cells. Treated cells show no incorporation of amino acid at 12.5 μM cycloheximide, indicating no new proteins are being translated. Experiment performed in three independent replicates ($n = 3$). (B) Quantification plot of phosphor plate results. Error bars are SEM ($n = 3$).

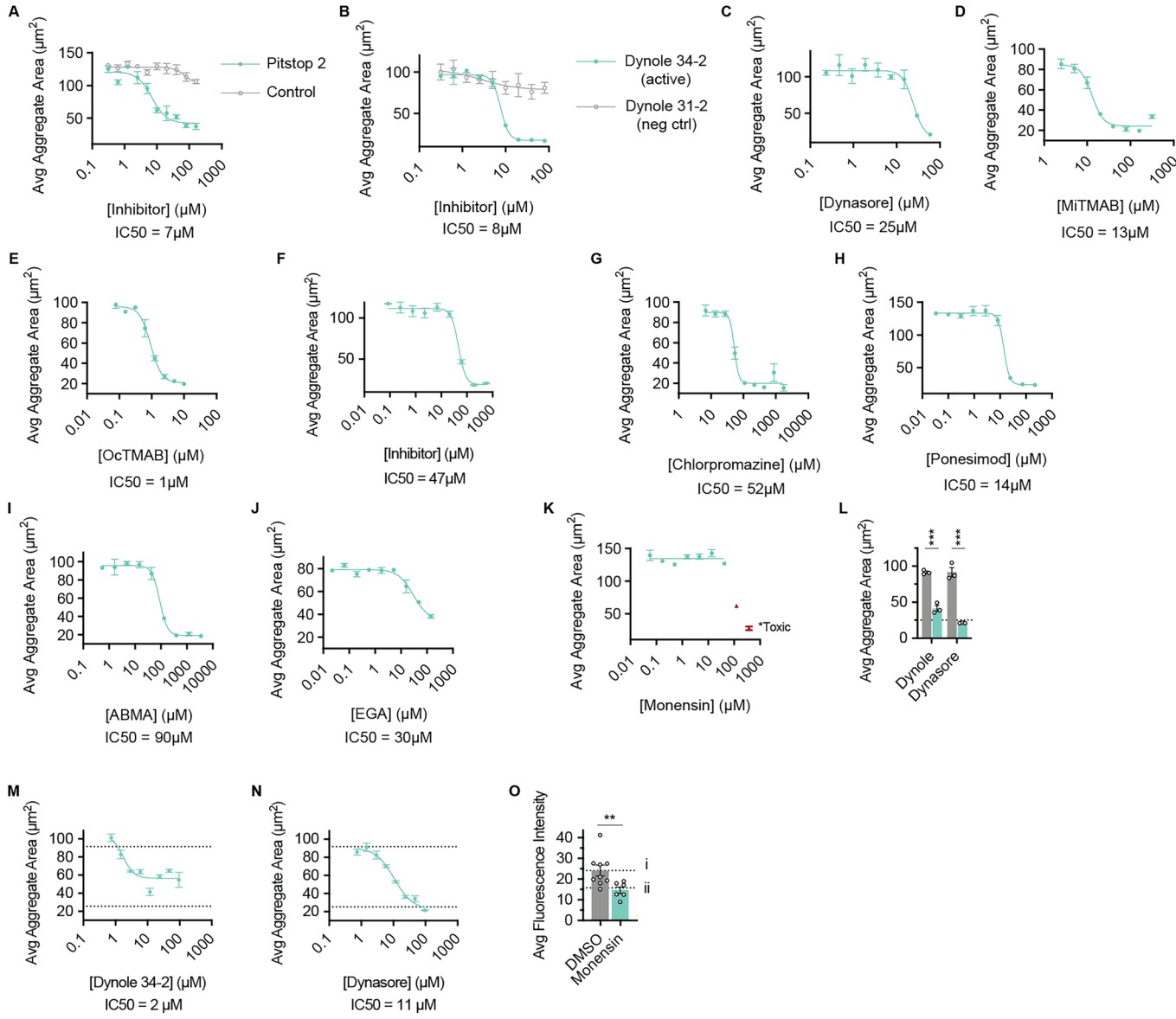

**Figure EV5. Individual dose–response curves for all endocytosis inhibitors.**

(**A–N**) Average aggregate areas resulting from a dilution series of various endocytosis inhibitors. (**A–K**) Aggregation was induced with 5% (v/v) FBS after 30 min of treatment with each inhibitor. (**A**) The clathrin recruitment inhibitor, Pitstop 2 inhibited aggregation while its negative control molecule did not. The $IC_{50}$ of Pitstop in Capsaspora was found to be 7 μM. The reported IC50 for Pitstop is reported to be 12 μM in HeLa cells (von Kleist et al, 2011). (**B**) The dynamin inhibitor Dynole 34-2 (Hill et al, 2009) inhibited aggregation while its negative control molecule Dynole 31-2 did not. (**C**) The dynamin inhibitor Dynasore (Kirchhausen et al, 2008) inhibited aggregation, although not as strongly as Dynole 34-2. (**D, E**) The dynamin recruitment inhibitors MiTMAB and OcTMAB (Quan et al, 2007) strongly inhibited aggregation. (**F**) The actin polymerization inhibitor CK-666 (Hetrick et al, 2013) prevented aggregation. (**G**) The clathrin decoating inhibitor chlorpromazine (Vercauteren et al, 2010) inhibited aggregation. (**H**) The endosome maturation inhibitor Ponesimod (Fauzyah et al, 2021) inhibited aggregation. (**I**) The endosome maturation inhibitor ABMA (Wu et al, 2017) inhibited aggregation. (**J**) EGA even at the highest concentration of solubility still had many visible aggregates (although looser) and so was considered not active. The $IC_{50}$ of EGA is reported to be 1 μM in A549 cells (Gillespie et al, 2013). (**K**) The lysosome pH acidification inhibitor Monensin (Misinzo et al, 2008) did not inhibit aggregation. (**L–N**) Aggregation was induced with 100 μg/mL POPC after 30 min of treatment with each inhibitor. Dashed lines indicate controls of 'no inhibitor' at the top and 'no added POPC' at the bottom. Panel (**L**) shows the most potent concentrations of the inhibitors (12 μM Dynole and 96 μM Dynasore), and panels (**M–N**) show all tested concentrations. (**O**) Average fluorescence intensity per cell of pHrodo red LDL after 30 min of treatment with the lysosome pH acidification inhibitor Monensin. Label (i) is the average intensity of cells with pHrodo staining in the absence of Monensin. Label (ii) is the average baseline fluorescence that appears as pHrodo staining in the absence of pHrodo addition. In all plots, error bars are SEM ($n = 3$). (**L, O**) Individual biological replicates are displayed with white circles, and each treatment was compared with its own untreated control by a $t$ test. $P$ values for each comparison follow: in (**L**), Dynole (0.0004), Dynasore (0.0004), and in (**O**) (0.0094).

