## [Peer Review File · EMBO Reports]

Endocytosed lipids induce cell aggregation via filopodia retraction in a close relative of animals

Ria Kidner, Eleanor Goldstone, Henry Rodefeld, Lorin Brokaw, Aria Gonzalez, Lalitha Sastry, Ranojoy Baisya, Núria Ros-Rocher, and Joseph Gerdt

Corresponding author(s): Joseph Gerdt (jpgerdt@iu.edu)

Review Timeline:

Submission Date:	10th Sep 25
Editorial Decision:	27th Oct 25
Revision Received:	10th Feb 26
Editorial Decision:	4th Mar 26
Revision Received:	10th Mar 26
Accepted:	18th Mar 26

Editor: Yehu Moran

Transaction Report:

Dear Dr. Gerdt

Thank you for the submission of your manuscript to EMBO reports. We have now received the full set of referee reports that are all pasted below.

As you will see, the referees acknowledge that the findings are interesting and valuable for those working in the field. However, they also raise some issues that require your attention.

I would thus like to invite you to revise your manuscript with the understanding that the referee concerns should be fully addressed and their suggestions taken on board whenever possible. Please address all referee concerns in a complete point-by-point response. Acceptance of the manuscript will depend on a positive outcome of a second round of review. It is EMBO reports policy to allow a single round of major revision only and acceptance or rejection of the manuscript will therefore depend quite heavily on the completeness of your responses included in the next, final version of the manuscript.

We realize that it is difficult to revise to a specific deadline. In the interest of protecting the conceptual advance provided by the work, we recommend a revision within 3 months (27th Jan 2026). Please discuss the revision progress ahead of this time with the editor if you require more time to complete the revisions.

- 1) A data availability section providing access to data deposited in public databases is missing. If you have not deposited any data, please add a sentence to the data availability section that explains that.
- 2) Your manuscript contains statistics and error bars based on $n=2$. Please use scatter blots in these cases. No statistics should be calculated if $n=2$.

<<https://www.embopress.org/page/journal/14693178/authorguide#expandedview>>

5) a complete author checklist, which you can download from our author guidelines

<<https://www.embopress.org/page/journal/14693178/authorguide>>. Please insert information in the checklist that is also reflected in the manuscript. The completed author checklist will also be part of the RPF.

6) Please note that all corresponding authors are required to supply an ORCID ID for their name upon submission of a revised manuscript (<<https://orcid.org/>>). Please find instructions on how to link your ORCID ID to your account in our manuscript tracking system in our Author guidelines

<<https://www.embopress.org/page/journal/14693178/authorguide#authorshipguidelines>>

10) Regarding data quantification (see Figure Legends:

<https://www.embopress.org/page/journal/14693178/authorguide#figureformat>)

12) All Materials and Methods need to be described in the main text using our 'Structured Methods' format, which is required for all research articles. According to this format, the Methods section includes a Reagents and Tools Table (listing key reagents, experimental models, software and relevant equipment and including their sources and relevant identifiers) followed by a Methods and Protocols section describing the methods using a step-by-step protocol format. The aim is to facilitate adoption of the methodologies across labs. More information on how to adhere to this format as well as a downloadable template (.docx) for the Reagents and Tools Table can be found in our author guidelines:

An example of a Method paper with Structured Methods can be found here: <https://www.embopress.org/doi/full/10.1038/s44320-024-00037-6#sec-4>

I look forward to seeing a revised form of your manuscript when it is ready.

Yours sincerely,

Yehu Moran
Academic Editor
EMBO Reports

Referee #1:

Kidner et al. present novel data on the chemical and physical nature of cell aggregation in the filasterean *Capsaspora owczarzaki*. This is a well written and well executed study that I thoroughly enjoyed reading and catching up on what has gone on in this emerging model organism over the last few years. I do have a few minor questions and comments below the authors should address and consider prior to the publication of the manuscript.

"Capsaspora likely inhabits vector snails to interfere with schistosome maturation and transmission." Do the authors believe this is really the reason *Capsaspora* inhabits vector snails. I think some rewording here would be appropriate. Do the authors have any explanation for the trend seen in Fig. S1A? That is why avg. aggregate size increases at 10min of lipid sonification then decreases for the 20, 30, and 40 min intervals before increasing again at 45min, followed by a slight decrease at 60min. I may have missed it but if not present please add what time point were the samples prepped for SEM taken from? "actin polymerization and branching with CK-666 (necessary for endosome budding into the clathrin pit)", prevented aggregation." Could the authors clarify for me why they believe this result is linked specifically to this endocytosis pathway? Wouldn't preventing actin polymerization prevent overall cell motility and therefore aggregation? Figure 5. I would strongly suggest not having the same color arrows represent different things in the same overall figure. I somewhat question the proposed model shown in figure 6. Do the authors have any evidence that cells in aggregates are feeding? I didn't gather that aggregation was involved in feeding in the paper that presented the original TEM data feeding in *Capsaspora*. In the presence of sporocysts alone this appears to be the case

Minor:

Page 1 Is a powerful protist -> is a protist

Page 1 Please add (hereafter "*Capsaspora*") in the first sentence of the introduction if *C. owczarzaki* is to simply be referred to by the genus name throughout the rest of the manuscript.

Movie caption 3. Typo "They were they"

Figure S3 caption 12um is written in the caption with the figure shows 12.5um. Which is correct?

Movie 2 caption. I do not see the time displayed in the top left corner of the video as described. The frame rate of the video as well as other time lapse movies would be a welcome addition.

Figure 6 caption. Reference listed as number 241 is a typo.

Referee #2:

Summary:

In this paper, Kidner et al. characterized the specificity and cell biology of lipid-induced aggregation in the Filasterean *Capsaspora owczarzaki*. The realization that lipid particles were necessary for signaling, as shown by electron microscopy (Figure S1) and the transition temperature of the most active lipids (Figure 1 K), compelled further cell biological study. The authors adduced that endocytic trafficking enabled the internalization of lipid signals. They first showed that aggregation does not require ongoing translation (cycloheximide inhibition in Figure 2 that was well supported by Figure S2 showing the direct impact of cycloheximide on translation). The authors then characterized 12 different chemical inhibitors of different stages of endocytosis, nicely performing dose-response curves for the effect of the drugs on aggregation (Figures 3 and S4). These data, fascinatingly indicate that lipids act to induce aggregation through endosomes before those vesicles become acidified. The authors then perform live-cell microscopy to further show the internalization of those vesicles. Together these experiments

provide evidence for the cell biological transformations that contribute to the early stages of aggregative multicellularity in *C. owczarazki*. These results have fascinating and important implications for the evolution and origin of animal multicellularity given the phylogenetic position of *C. owczarazki* in the sister group of choanozoans. Overall, the study is sound and significant, and the authors have ample evidence for their main conclusions. The comments below serve to more clearly show those conclusions. With those changes, this paper should be published.

Major Comments:

1. Figure 2: Were any of the vital dyes co-administered to directly compare lipid trafficking to localization of acidic vesicles (pHrodo)? Co-localization would support the point that lipids traffic to compartments other than acidic vacuoles to induce aggregation.
2. For chloroquine and monensin, do the authors have supporting assays to show that these chemicals do act as expected (e.g. chloroquine reduction in pHrodo staining)? Without these data, aggregation in the presence of these drugs (Figure 2) could simply be due to the lack of activity of these drugs rather than support the conclusion that signaling occurs upstream of vesicle acidification. In Figure S4K, the authors only show the effect of these drugs on cell aggregation.
3. Please perform experiments in Figure 5 with some of the inhibitors characterized in Figure 3 to support that the endocytic pathway internalizes vesicles. This experiment will also support the conclusion that those inhibitors block aggregation through their inhibition of endocytosis and not by an unknown mechanism.
4. The experiments in Figure 5 use common laser and filter to visualize the cellular membrane and DOPC particles. The combination of two labeled molecules in one channel detracts adds caveats to conclusions about ingestion and trafficking- perhaps some enrichments of membrane signal is due to other, unaccounted factors. I do agree with the authors' conclusions as the most plausible, but it would be nice to see these the cellular membrane labels in a different color than the lipid particles to for a cleaner experiment. The authors do mention some of these alternative explanations in the discussion ("Since the inducer is a lipid, it may alternatively...")
5. I cannot easily tell how many times the experiments in Figures 4 & 5 were replicated. There are no quantifications of trafficking rates from different experiments. Replicate images from independent experiments for Figures 4 & 5 should at least be shown in the supplementary materials, and the authors should provide that information regarding replicates in the figure legends and materials & methods (e.g. images are representative of N independent experiments with M cells characterized in each experiment)

Minor Comments:

1. The authors should mention the concentration of inhibitors in the main text and/or figure legends, such as for Figure 3G.
2. Please spell-out the abbreviation for NMM-Venus when first mentioned on p. 14.
3. The discussion could be enriched by discussing how proteins that are specifically phosphorylated during aggregation could contribute to enhanced trafficking and cellular projections.

Referee #3:

In this report Kidner et al. build on the Gerdt lab's previous work showing that extracellular lipids regulate multicellular aggregation in the filasterian *Capsaspora*, a close unicellular relative of animals. Given this close relatedness, and the fact that another filasterian has recently been reported to similarly aggregate, characterizing this mode of regulation could inform our understanding of the emergence of animal multicellularity and thus is of significant interest. The group previously found that lipids in laboratory culture medium (from serum LDLs) and lipids present in the putative snail hosts of *Capsaspora* (phosphatidylcholines) can induce aggregation. Here they attempt to characterize the specificity of the lipid signal and nicely show that the signal is not a specific molecule, but instead is defined by particular structural criteria (a zwitterionic head group and two hydrophobic tails with at least one unsaturated bond). They further characterize the aggregation process, presenting evidence that aggregation induction involves contraction at intercellular filopodial contacts, and is independent of translation but dependent on clathrin-mediated endocytosis, as lipid vesicles appear to be endocytically taken up by filopodia and trafficked to the cell body, apparently triggering an aggregation response. The authors should be commended for such a detailed characterization of signaling in a non-model organism. The manuscript is clear and well written, and the data largely support the conclusions. However for some conclusions, such as the directional trafficking of endosomes through filopodia and the contractile filopodial behavior during aggregation, I feel a more rigorous analysis or a less strong conclusion is needed. My major and minor concerns in the manuscript are outlined below:

Major concerns

Throughout the manuscript aggregate area is quantified, with an average of about 100 square microns for a positive control. If a *Capsaspora* cell is about 5 microns across, this aggregate size would consist of something like 4 cells. However in the given pictures, aggregates appear much larger, apparently consisting of tens or hundreds of cells, which should have much larger area. The authors should address this apparent discrepancy- if these numbers are correct, then they should describe in their

quantification methods why aggregate size in the quantifications are so different from what might be expected by the images. Maybe a histogram of aggregate sizes would be useful.

In figure 5 A-B the authors argue that PC-containing vesicles are trafficked to the cell body based on time lapse recordings. Although it appears that the PC signal is moving toward the cell body, I don't think you can confidently conclude trafficking with the evidence presented. In movie 5 there appear to be at least a few examples of filopodia that are clearly stuck and not retracting, and the vesicles in these filopodia do not appear to be moving. To better evaluate this conclusion, the authors should quantify the displacement of vesicles relative to the distal end of the filopodia- this would control for apparent trafficking that is actually due to filopodial retraction. If this experiment is technically infeasible, for example due to filopodial ends not staying in the focal plane, the authors should be more careful in their conclusions.

Similarly, in figure 5 C-D the authors argue that during aggregation filopodia are retracted at cell contacts but not retracted when not in cell contact. In movie 8 there is a region in the top right where substrate-level filopodia are clearly visible and to my eye there appears to be retraction of filopodia even when there is not contact with another cell. The authors should quantify the distance from the filopodial end to the cell body of non-contacting cells, and the cell body to cell body distance of filopodial interacting cells during addition of PCs during some time interval, with controls where the PCs are not added. Again if this experiment is technically not possible due to visualizing the filopodia ends then the authors should be more reserved in their conclusions.

Minor concerns

Page 2:

"aggregation is initiated through post-translational activation of filopodial retraction" I think that "post-translational" generally refers to protein modifications and not cellular behaviors- something like "independent of protein translation" would be more clear.

Page 3:

"powerful protist of particular interest" I'm not sure what "powerful" is referring to in this sentence.

"For example, aggregation ... drives immune cell aggregation during inflammation" The end of this sentence should be rephrased.

Figure 1:

Although the conclusions in the manuscript are often evident from the graphs, I still think that some statistical analyses would be good to have in, for instance, 1E and 1H.

Page 8:

"To test this, we inhibited protein translation using cycloheximide treatment on single cells" Do the authors mean "non-aggregated cells" by single?

Figure 4:

I think this figure would benefit from clearer labeling- for instance in D the legend describes a "white" channel but there is no white signal in the images.

Figure 5:

In panels A and D do the blue and orange arrows indicate different things?

Page 19:

"remains to be seen whether *Capsaspora* filopodia retract by actomyosin contraction" There is a report that the myostatin inhibitor blebbistatin causes "loosening" of aggregates (Phillips and Pan *Elife* 2024). I think this fact may be worth mentioning in relation to the filopodial contraction findings.

Figure S2:

In panel E what does the star indicate?

In addition to lipid vesicles both gentle shaking and culturing in low adherence conditions have been reported to induce *Capsaspora* aggregation. Given these other two conditions, it raises the question of whether the lipid vesicle signals affect the cell-substrate adhesion strength of *Capsaspora* cells. Either here or in a follow up report it would be interesting to ask if the lipid vesicles reduce cell-substrate adhesion, thus biasing the cells to cell-cell adhesion and aggregation.

In movie 8 it appears that, even when only visualizing NMM-venus, more intracellular vesicles are visible in cells. Could the PC vesicles be generally increasing endocytosis? Can this be measured?

Reviewer 1:

We appreciate the reviewer's positive assessment that our study is well written and executed, and we are grateful for your constructive and detailed feedback.

1. The Reviewer states: "*Capsaspora likely inhabits vector snails to interfere with schistosome maturation and transmission.*" *Do the authors believe this is really the reason Capsaspora inhabits vector snails. I think some rewording here would be appropriate.'*

We thank the reviewer for pointing this text out. We did not mean to imply that the natural 'purpose' of *Capsaspora* is to exclude schistosomes. We have rephrased this text to now read:

"Moreover, *Capsaspora* responds to specific snail-derived lipids^{11, 21} by forming large aggregates (both in vitro and within snail tissue), suggesting that this behavior is important for host colonization. However, we still poorly understand the regulatory mechanisms by which *Capsaspora* senses these host-derived factors."

2. The Reviewer states: '*Do the author's have any explanation for the trend seen in Fig. S1A? That is why avg. aggregate size increases at 10min of lipid sonification then decreases for the 20, 30, and 40 min intervals before increasing again at 45min, followed by a slight decrease at 60min. I may have missed it but if not present please add what time point were the samples prepped for SEM taken from?*

We do not believe this trend is significant. The error bars indicate biological triplicates (performed on three independent sets of cells), but each of the three biological replicates was treated using the same vesicle preparations. We therefore think that the small variation observed likely reflects minor random variability in vesicle preparation effectiveness. We chose not to repeat the experiment with triplicate vesicle batches at each time point, as the current results show that 10 minutes is sufficient to generate active vesicles. This was the main goal of the experiment, and we used this condition for all subsequent experiments in the paper. To clarify, we added the following **bold** text to the figure caption:

"Error bars represent SEM of biological triplicates (n=3) **from a single batch of sonicated vesicles for each time point.**"

Regarding the TEM time points, we thank the reviewer for noting this and have added this detail to the methods and Fig. EV2 legend (former Fig. S1).

3. The Reviewer states: "*actin polymerization and branching with CK-666 (necessary for endosome budding into the clathrin pit), prevented aggregation.*" *Could the authors clarify for me why they believe this result is linked specifically to this endocytosis pathway? Wouldn't preventing actin polymerization prevent overall cell motility and therefore aggregation?'*

We agree that this inhibitor may have other effects on the cell that could influence aggregation, and we now acknowledged this caveat in the manuscript. Our strategy was to exhaustively test a broad set of inhibitors that target different components of the endocytic pathway, so as not to rely on a single (or even a small number of) pharmacological inhibitors that were developed for non-*Capsaspora* systems. While each inhibitor may carry its own caveats, taken together, they

show compelling evidence that endocytosis is important for aggregation. We have clarified this point in the revised text:

“Admittedly, each inhibitor has the potential for off-target effects that may inhibit aggregation apart from endocytosis blockade. Perhaps most notably, CK-666 may inhibit cell motility in a way that delays or prevents aggregation. However, the consistent inhibition of aggregation by this exhaustive set of inhibitors, each targeting a different component of the endocytosis pathway, suggests that endocytosis is required for aggregation.

4. The Reviewer states: *‘Figure 5. I would strongly suggest not having the same color arrows represent different things in the same overall figure.’*

We thank the review for the suggestion and have edited the colors to be unique.

5. The Reviewer states: *‘I somewhat question the proposed model shown in figure 6. Do the authors have any evidence that cells in aggregates are feeding? I didn’t gather that aggregation was involved in feeding in the paper that presented the original TEM data feeding in Capsaspora. In the presence of sporocysts alone this appears to be the case’*

We thank the reviewer for their insightful comment. We agree that direct evidence of feeding within *Capsaspora* aggregates has been limited in the literature. However, we have now incorporated references that provide visual support for this possibility. Specifically, both [DOI: 10.1016/0022-2011(79)90149-6] and [DOI: 10.1371/journal.ppat.1013440] show *Capsaspora* cells forming what appears to be aggregates coating schistosome prey. In the 1979 paper, this is visible Figures 8 & 10, and in the 2025 paper, this is shown in videos S1-S3.

While these observations do not constitute definitive proof that feeding occurs exclusively within aggregates, they do suggest that aggregation can accompany or facilitate interactions with prey. We have clarified this point in the manuscript to avoid overinterpretation while still acknowledging the available evidence:

“Modest evidence from the literature supports this link between aggregation and feeding, as *Capsaspora* has been shown to form apparent aggregates onto schistosome prey, and the aggregates disperse after the prey has been consumed.[refs]”

6. The Reviewer suggests minor edits:

Page 1 Is a powerful protist -> is a protist

We thank the reviewer and have incorporated the suggested edit. We have also edited the introduction to improve clarity

Page 1 Please add (hereafter "Capsaspora") in the first sentence of the introduction if C. owczarzaki is to simply to be referred to by the genus name throughout the rest of the manuscript.

We thank the reviewer and have incorporated the suggested edit.

Movie caption 3. Typo "They were they"

We thank the reviewer and have fixed the typo.

Figure S3 caption 12um is written in the caption with the figure shows 12.5um. Which is correct?

We thank the reviewer and have fixed the typo in the figure caption.

Movie 2 caption. I do not see the time displayed in the top left corner of the video as described. The frame rate of the video as well as other time lapse movies would be a welcome addition.

We thank the reviewer for pointing this out. We see the time displayed, but it is a smaller font than the other videos. We increased the font size to be clearer. However, we noticed that it still can be hidden behind controls depending on the video playback software used.

Figure 6 caption. Reference listed as number 241 is a typo.

We thank the reviewer and have fixed the typo (should have read 2, 41, 42).

Reviewer 2:

We thank the reviewer for your positive and enthusiastic comments and for your constructive feedback that significantly improved the manuscript.

1. The Reviewer states: *'Figure 2: Were any of the vital dyes co-administered to directly compare lipid trafficking to localization of acidic vesicles (pHrodo)? Co-localization would support the point that lipids traffic to compartments other than acidic vacuoles to induce aggregation.'*

We believe this question refers to Figure 3. No, we have not co-administered the fluorescent PC vesicles with pHrodo or any other dyes. We would like to clarify that we do not propose that the vesicles need to get far along the endocytic pathway into an acidic compartment in order to induce aggregation (see below). Likewise, we do not interpret the lack of impact by monensin to mean that these vesicles are going to other places besides acidic vacuoles.

Our data are consistent with vesicles reaching acidic vacuoles – likely for nutrient processing – just as LDLs do. However, it also suggests that the induction of aggregation does not require this full extent of trafficking. The endocytosed LDLs and lipid vesicles also trigger aggregation, but the full trafficking to acidic vacuoles is not necessary for inducing aggregation. We have revised the text to clarify this interpretation, that now reads:

“The endocytosed LDLs (and presumably also the endocytosed PCs) ultimately accumulate in acidified lysosomes, as shown by activation of pHrodo dyes. However, our data suggest that the late stages of lysosome maturation are not required for initiating aggregation. Instead, only the early and intermediate steps of the endocytic pathway appear to be required to induce *Capsaspora* aggregation.”

2. The Reviewer states: *'For chloroquine and monensin, do the authors have supporting assays to show that these chemicals do act as expected (e.g. chloroquine reduction in pHrodo staining)? Without these data, aggregation in the presence of these drugs (Figure 2) could simply be due to the lack of activity of these drugs rather than support the conclusion that signaling occurs upstream of vesicle acidification. In Figure S4K, the authors only show the effect of these drugs on cell aggregation.'*

We agree that this point was not sufficiently evident in our earlier figures and required more extensive analysis. We have now run these experiments and found that actually chloroquine surprisingly did not significantly inhibit pHrodo staining at the tested concentrations. Therefore, we removed all chloroquine experiments from the paper and we sincerely thank the reviewer for this suggestion! Monensin, on the other hand, did inhibit pHrodo staining, confirming its ability to inhibit endosome acidification in *Capsaspora*. The monensin data have been added to **Fig EV50**, and the following **bold** text was added:

“Moreso, inhibition of lysosome acidification using **Monensin failed to prevent aggregation (even though it did inhibit the formation of acidified endosomal compartments, Fig. EV50).”**

3. The Reviewer states: *'Please perform experiments in Figure 5 with some of the inhibitors characterized in Figure 3 to support that the endocytic pathway internalizes vesicles. This experiment will also support the conclusion that those inhibitors block aggregation through their inhibition of endocytosis and not by an unknown mechanism.'*

We thank the reviewer for their suggestion. We have now run the aggregation inhibition experiments of Figure 3G using POPC induction. We used two well-validated inhibitors Dynole & Dynasore. These data have been added to **Fig. EV5L-N**. We have also run the imaging experiment of Fig. 5A with Dynole, which showed no PC uptake or aggregation induction. These videos are attached as new **Movies EV10–11 (see Movies EV7-9 as reference for untreated cells)**.

4. The Reviewer states: *'The experiments in Figure 5 use common laser and filter to visualize the cellular membrane and DOPC particles. The combination of two labeled molecules in one channel detracts adds caveats to conclusions about ingestion and trafficking-perhaps some enrichments of membrane signal is dues to other, unaccounted factors. I do agree with the authors' conclusions as the most plausible, but it would be nice to see these the cellular membrane labels in a different color than the lipid particles to for a cleaner experiment. The authors do mention some of these alternative explanations in the discussion ("Since the inducer is a lipid, it may alternatively...")'*

We agree that having both labels in one channel is not ideal. However, our efforts to clearly label filopodia or vesicles in a red channel have failed. Our NMM-mScarlet line is not bright enough to show clear filopodia. Efforts to stain membranes chemically (e.g., FM4-64, MemGlow 488) have also failed because *Capsaspora* quickly endocytoses these membrane stains. We also have not yet been able to obtain a non-green fluorescent lipid that stably labels the vesicles. We have further expanded on this limitation in the text (see below, bold parts are new), along with edits to address Reviewer 3's comment 2. We also added the word "hypothesis" to Figure 5B to emphasize the need for further validating experiments.

"Although the membrane and PC fluorescence channels overlapped, we observed **instances of bright PC puncta appearing to traffic** along extended, non-retracting filopodia into the cell bodies. **Furthermore, this uptake and trafficking was prevented by the endocytosis inhibitor Dynole (Movies EV10–11). However, this trafficking was not universal: many vesicles appeared to be stuck on filopodia and failed to traffic (Movies EV6–9). Since the event sometimes occurs, we hypothesize that Capsaspora can actively traffic exogenously added PC vesicles along outstretched filopodia (Fig. 5B), but yet-unknown factors may determine the likelihood of this process. Improved imaging that maintains filopodia in the focal plane and labels filopodia and lipids in separate channels will enable robust and quantitative assessments of this putative trafficking.**"

5. The Reviewer states: *'I cannot easily tell how many times the experiments in Figures 4 & 5 were replicated. There are no quantifications of trafficking rates from different experiments. Replicate images from independent experiments for Figures 4 & 5 should at least be shown in the supplementary materials, and the authors should provide that information regarding replicates in the figure legends and materials & methods (e.g. images are representative of N independent experiments with M cells characterized in each experiment)'*

We have now included a triplicate set of videos for each experiment in figures 4 and 5, and we have referenced the videos in the text and figure captions. As mentioned below in response to reviewer 3, due to technical limitations, we have not rigorously quantified trafficking and so we now make that clear with additional text below.

“Improved imaging that maintains filopodia in the focal plane and labels filopodia and lipids in separate channels will enable robust and quantitative assessments of this putative trafficking.”

6. The Reviewer suggests minor edits:

The authors should mention the concentration of inhibitors in the main text and/or figure legends, such as for Figure 3G.

We thank the reviewer and have added the concentrations to the figure legend.

Please spell-out the abbreviation for NMM-Venus when first mentioned on p. 14.

We thank the reviewer and have incorporated the suggested edit.

The discussion could be enriched by discussing how proteins that are specifically phosphorylated during aggregation could contribute to enhanced trafficking and cellular projections.

We thank the reviewer and have added new discussion of those results and a new reference [DOI: 10.1016/j.devcel.2016.09.019].

“The quick filopodial retraction observed here suggests a possible role for protein phosphorylation to regulate aggregation. Previous work reported that genes involved in regulating the actin cytoskeleton have many phosphosites, suggesting a complex phosphoregulation of these genes.[ref] Their work further reported differential phosphorylation of the “actin cytoskeleton reorganization” gene ontology term between aggregates and non-aggregated cells. Therefore, we hypothesize that a change in protein phosphorylation leads to the rapid retraction of actin-filled filopodia. This hypothesis requires more thorough testing under our rapid chemically induced aggregation conditions.”

Reviewer 3:

We thank the reviewer for their positive assessment of our work and for the helpful comments on how to clarify and improve our interpretations.

1. The Reviewer states: *‘Throughout the manuscript aggregate area is quantified, with an average of about 100 square microns for a positive control. If a Capsaspora cell is about 5 microns across, this aggregate size would consist of something like 4 cells. However in the given pictures, aggregates appear much larger, apparently consisting of tens or hundreds of cells, which should have much larger area. The authors should address this apparent discrepancy- if these numbers are correct, then they should describe in their quantification methods why aggregate size in the quantifications are so different from what might be expected by the images. Maybe a histogram of aggregate sizes would be useful.’*

We thank the reviewer for raising this point, which will indeed help prevent potential confusion for readers. Even in experiments with strong aggregation responses, we consistently observe some heterogeneity: most cells form aggregates (which can fuse and grow larger), but a fraction remains as single cells. Our analysis pipeline quantifies all detected binarized objects – both single cells and aggregates – as individual particles. As a result, if an image contains, for example, 10 large aggregates and 10 single cells, the calculated ‘average aggregate area’ for that image will be approximately half the true area of the aggregates. Therefore, this metric reflects the *overall degree of aggregation in the well* (which remains unchanged in control wells lacking aggregation), rather than the size of individual aggregates. We have clarified this in the caption for Fig. 1 and, as suggested, added a representative histogram showing the distribution of particle sizes as new Fig. EV1.

2. The Reviewer states: *‘In figure 5 A-B the authors argue that PC-containing vesicles are trafficked to the cell body based on time lapse recordings. Although it appears that the PC signal is moving toward the cell body, I don't think you can confidently conclude trafficking with the evidence presented. In movie 5 there appear to be at least a few examples of filopodia that are clearly stuck and not retracting, and the vesicles in these filopodia do not appear to be moving. To better evaluate this conclusion, the authors should quantify the displacement of vesicles relative to the distal end of the filopodia- this would control for apparent trafficking that is actually due to filopodial retraction. If this experiment is technically infeasible, for example due to filopodial ends not staying in the focal plane, the authors should be more careful in their conclusions.’*

We appreciate the concern of the reviewer. Indeed, the filopodia move in and out of the focal plane, making this measurement very challenging. Also, it would be much better to do with the vesicles and filopodia labeled in different channels. We are working to get this, but it will take too long and be well-suited for a follow-up manuscript that focuses on vesicle trafficking. Therefore, we have tempered our conclusions in this paper. See the new text below with changes in bold (also related to Reviewer 2’s comment 4). We also added the word “hypothesis” to Figure 5B to emphasize the need for further validating experiments.

“Although the membrane and PC fluorescence channels overlapped, we observed **instances of** bright PC puncta **appearing to traffic** along extended, non-retracting filopodia into the cell bodies. **Furthermore, this uptake and trafficking was prevented by the endocytosis inhibitor Dynole (Movies EV10–11).** However, this trafficking was not universal: many vesicles appeared to be stuck on filopodia and failed to traffic (Movies EV6–9). Since the event sometimes occurs, we hypothesize that

***Capsaspora* can actively traffic exogenously added PC vesicles along outstretched filopodia (Fig. 5B), but yet-unknown factors may determine the likelihood of this process. Improved imaging that maintains filopodia in the focal plane and labels filopodia and lipids in separate channels will enable robust and quantitative assessments of this putative trafficking.**

Additionally, we are currently working to improve this imaging using multiple labels, and we expect to provide more comprehensive quantitative trafficking data in future work.

3. The Reviewer states: *‘Similarly, in figure 5 C-D the authors argue that during aggregation filopodia are retracted at cell contacts but not retracted when not in cell contact. In movie 8 there is a region in the top right where substrate-level filopodia are clearly visible and to my eye there appears to be retraction of filopodia even when there is not contact with another cell. The authors should quantify the distance from the filopodial end to the cell body of non-contacting cells, and the cell body to cell body distance of filopodial interacting cells during addition of PCs during some time interval, with controls where the PCs are not added. Again if this experiment is technically not possible due to visualizing the filopodia ends then the authors should be more reserved in their conclusions.’*

We appreciate the concern of the reviewer and have tempered our conclusions accordingly. We do not mean to imply that filopodia cannot move in the absence of cell-cell contacts. Indeed, as the reviewer notes, filopodia are highly dynamic in non-aggregating cells. They can retract and extend or stay in place. We have now clarified in the text that in single aggregating cells, not every filopodium retracts synchronously. In other words, there is not a uniform cellular signal that retracts all filopodia in that cell. Rather, some filopodia retract and draw the cells together, and other filopodia remain outstretched. See new text including reference [DOI: 10.1016/j.cub.2020.08.015] added to document:

“Therefore, the PC lipids did not uniformly induce retraction of all filopodia throughout the cell. This observation suggests that PCs may not be sufficient to induce filopodia retraction, but retraction may also require cell-cell contact by a filopodia as a *second ‘signal’* for that filopodium to retract (Fig. 5E). Filopodia are highly dynamic cellular components that are also used for substrate adhesion and motility;[ref] therefore, PCs may have important influences on these filopodia-driven behaviors, as well.”

4. The Reviewer suggests minor edits:

Page 2: "aggregation is initiated through post-translational activation of filopodial retraction" I think that "post-translational" generally refers to protein modifications and not cellular behaviors- something like "independent of protein translation" would be more clear.

We thank the reviewer and have incorporated the suggested edit.

Page 3: "powerful protist of particular interest" I'm not sure what "powerful" is referring to in this sentence.

We thank the reviewer and have removed ‘powerful’.

Page 3: "For example, aggregation ... drives immune cell aggregation during inflammation" The end of this sentence should be rephrased.

We thank the reviewer and have rephrased the sentence to avoid the repetition of 'aggregation'.

Figure 1: Although the conclusions in the manuscript are often evident from the graphs, I still think that some statistical analyses would be good to have in, for instance, 1E and 1H.

We thank the reviewer and have added statistical analysis to Fig. 1B,E,G & H and in Fig. 3G.

Page 8: "To test this, we inhibited protein translation using cycloheximide treatment on single cells" Do the authors mean "non-aggregated cells" by single?

Yes, we thank the reviewer for pointing out the confusion and have rephrased this text.

Figure 4: I think this figure would benefit from clearer labeling- for instance in D the legend describes a "white" channel but there is no white signal in the images.

We thank the reviewer and have added labels to the figures themselves and also clarified in the text and legend that no white signal is visible because these vesicles were not taken up.

Figure 5: In panels A and D do the blue and orange arrows indicate different things?

We thank the reviewer and have re-colored the arrows and explained what they indicate. In response also to reviewer 2, we have changed the colors in panel D to not conflict with colors in panel A. We made retracting filopodia that are connected between cells be one color and filopodia that remain outstretched be a different color. This is explained in the figure caption.

Page 19: "remains to be seen whether Capsaspora filopodia retract by actomyosin contraction" There is a report that the myostatin inhibitor blebbistatin causes "loosening" of aggregates (Phillips and Pan Elife 2024). I think this fact may be worth mentioning in relation to the filopodial contraction findings.

Indeed, we thank the reviewer for this suggestion and have added this published finding to our text.

Figure S2: In panel E what does the star indicate?

We thank the reviewer for catching this oversight. We added the explanation to the caption. The cells formed a single giant aggregate in two of the replicates, which caused our quantification method to provide a low "average aggregate area." See also our response to Reviewer 3's first question above.

In addition to lipid vesicles both gentle shaking and culturing in low adherence conditions have been reported to induce Capsaspora aggregation. Given these other two conditions, it raises the question of whether the lipid vesicle signals affect the cell-substrate adhesion strength of Capsaspora cells. Either here or in a follow up report it would be interesting to ask if the lipid vesicles reduce cell-substrate adhesion, thus biasing the cells to cell-cell adhesion and aggregation.

We have previously shown that lipid inducers are required to induce aggregation [DOI: 10.1073/pnas.2216668120]. Shaking alone or low adherence conditions alone cannot induce aggregation without LDL/HDL/PC lipids. Nonetheless, we agree it is a great point to investigate how these lipids may influence adherence of cells to the substrate. It is certainly worth exploring in the future.

In movie 8 it appears that, even when only visualizing NMM-venus, more intracellular vesicles are visible in cells. Could the PC vesicles be generally increasing endocytosis? Can this be measured?

Indeed, we have noticed this too. We hypothesize that the NMM-fluorescent protein is incorporated into regions of the plasma membrane that bud to form the endocytic vesicles. We also hypothesize that the presence of PCs triggers their endocytosis (possibly via a receptor-mediated process), which in turn increases the overall level of endocytosis occurring in the cell. Quantifying this effect more rigorously is an excellent suggestion, and we plan to address it in future work.

Dear Dr. Gerdt

Thank you for the submission of your revised manuscript to our offices. I have went carefully over your revision (EMBOR-2025-62734V2) and in principle I am happy to accept in principle your paper to EMBO Reports, pending some essential technical corrections detailed below.

I look forward to seeing a new revised version of your manuscript as soon as possible.

Best regards,

Yehu Moran
Academic Editor
EMBO Reports

****specific comments by our editorial assistance team****

Conflict of Interest/DCIS: in, but it needs to be renamed to Disclosure and Competing Interests Statement and provided after Acknowledgments. Please correct.

Author Contributions/CRedit: need to be removed from the manuscript text and inserted only via the submission system.

REFERENCES: NOT OK - need to be alphabetical, not numerical; et al needs to be used after 10 author names; DOIs should only be used for preprints and datasets that have not been published yet. Please correct.

FUNDING INFO: not congruent between submission system and manuscript text as the Institut Pasteur is missing from the system as a separate entry. Please correct.

MOVIES: 16 movies uploaded; movies need to be uploaded as separate movie zip folders, each having a separate text file for the legend and the other one is the movie file; the majority of movies have been correctly uploaded but there are some that are combined with the others and need to be separated.

SYNOPSIS IMAGE: missing, please provide. It is recommended to look on the instructions for authors as well as recently published papers to get a better idea of the format.

SYNOPSIS TEXT: missing, please provide. It is recommended to look on the instructions for authors as well as recently published papers to get a better idea of the format.

R&T TABLE: currently in the manuscript text, needs to be removed from the text and uploaded separately.

Additional NOTES:

- MATERIALS & CORRESPONDENCE section should be removed from the text
- MATERIALS AND METHODS should be renamed METHODS

FIGURE CHECK:

Figure Legends - Comments

- Please note that the exact p values are not provided in the legends of figures 3G, EV5 L, O. This has to be fixed.

Data availability: please note that all relevant data should become publicly available to allow final acceptance.

The authors have addressed all minor editorial requests.

Joseph Gerdt
Indiana University
Department of Chemistry
United States

Dear Dr. Gerdt,

I am very pleased to accept your manuscript for publication in the next available issue of EMBO reports. Thank you for your contribution to our journal.

You may qualify for financial assistance for your publication charges - either via a Springer Nature fully open access agreement or an EMBO initiative. Check your eligibility: <https://link.springer.com/journal/44319/how-to-publish-with-us>

Yours sincerely,

Yehu Moran
Academic Editor
EMBO Reports

>>> Please note that it is EMBO Reports policy for the transcript of the editorial process (containing referee reports and your response letter) to be published as an online supplement to each paper. If you do NOT want this, you will need to inform the Editorial Office via email immediately. More information is available here: <https://link.springer.com/partners/embo-press/editorial-policies#Peer%20review>